# Dirac, Majorana, Weyl in 4D

**Loriano Bonora** [1,2,*], **Roberto Soldati** [3] **and Stav Zalel** [4]

1   International School for Advanced Studies (SISSA), via Bonomea 265, 34136 Trieste, Italy
2   INFN-Sezione di Trieste, Padriciano, 99, 34149 Trieste TS, Italy
3   Dipartimento di Fisica e Astronomia, via Irnerio 46, 40126 Bologna, Italy; roberto.soldati@bo.infn.it
4   Blackett Laboratory, Imperial College, London SW7 2AZ, UK; stav.zalel11@imperial.ac.uk
*   Correspondence: bonora@sissa.it

**Abstract:** This is a review of some elementary properties of Dirac, Weyl and Majorana spinors in 4D. We focus in particular on the differences between massless Dirac and Majorana fermions, on one side, and Weyl fermions, on the other. We review in detail the definition of their effective actions, when coupled to (vector and axial) gauge fields, and revisit the corresponding anomalies using the Feynman diagram method with different regularisations. Among various well known results we stress in particular the regularisation independence in perturbative approaches, while not all the regularisations fit the non-perturbative ones. As for anomalies, we highlight in particular one perhaps not so well known feature: the rigid relation between chiral and trace anomalies.

**Keywords:** spinors in 4d; regularization; anomalies

## 1. Introduction

This paper is a review concerning the properties of Dirac, Weyl and Majorana fermions in a 4 dimensional Minkowski space-time. Fermions are unintuitive objects, thus the more fascinating. The relevant literature is enormous. Still problems that seem to be well understood, when carefully put under scrutiny, reveal sometimes unexpected aspects. The motivation for this paper is the observation that while Dirac fermions are very well known, both from the classical and the quantum points of view, Weyl and Majorana fermions are often treated as poor relatives [1] of the former, and, consequently, not sufficiently studied, especially for what concerns their quantum aspects. The truth is that these three types of fermions, while similar in certain respects, behave radically differently in others. Dirac spinors belong to a reducible representation of the Lorentz group, which can be irreducibly decomposed in two different ways: the first in eigenstates of the charge conjugation operator (Majorana), the second in eigenstates of the chirality operator (Weyl). Weyl spinors are bound to preserve chirality, therefore do not admit a mass term in the action and are strictly massless. Dirac and Majorana fermions can be massive. In this review we will focus mostly on massless fermions, and one of the issues we wish to elaborate on is the difference between massless Dirac, Majorana and Weyl fermions.

The key problem one immediately encounters is the construction of the effective action of these fermions coupled to gauge or gravity potentials. Formally, the effective action is the product of the eigenvalues of the relevant kinetic operator. The actual calculus can be carried out either perturbatively or non-perturbatively. In the first case the main approach is by Feynman diagrams, in the other case by analytical methods, variously called Seeley–Schwinger–DeWitt or heat kernel methods. While

---

1   Actually, the Weyl spinors are the main bricks of the original Standard Model, while the neutral Majorana spinors could probably be the basic particle bricks of Dark Matter, if any.

the procedure is rather straightforward in the Dirac (and Majorana) case, the same approach in the Weyl case is strictly speaking inaccessible. In this case, one has to resort to a roundabout method, the discussion of which is one of the relevant topics of this review. In order to clarify some basic concepts we carry out a few elementary Feynman diagram calculations with different regularisations (mostly Pauli–Villars and dimensional regularisation). The purpose is to justify the methods used to compute the Weyl effective action. A side bonus of this discussion is a clarification concerning the nonperturbative methods and the Pauli–Villars (PV) regularisation: contrary to the dimensional regularisation, the PV regularisation is unfit to be extended to the heat kernel-like methods, unless one is unwisely willing to violate locality.

A second major ground on which Weyl fermions split from Dirac and Majorana fermions is the issue of anomalies. To illustrate it in a complete and exhaustive way we limit ourselves here to fermion theories coupled to external gauge potentials and, using the Feynman diagrams, we compute all the anomalies (trace and gauge) in such a background. These anomalies have been calculated elsewhere in the literature in manifold ways and since a long time, so that there is nothing new in our procedure. Our goal here is to give a panoptic view of these computations and their interrelations. The result is interesting. Not only does one get a clear vantage point on the difference between Dirac and Weyl anomalies, but, for instance, it transpires that the rigid link between chiral and trace anomalies is not a characteristic of supersymmetric theories alone, but holds in general.

The paper is organized as follows. Section 2 is devoted to basic definitions and properties of Dirac, Weyl and Majorana fermions, in particular to the differences between massless Majorana and Weyl fermions. In Section 3, we discuss the problem related to the definition of a functional integral for Weyl fermions. In Section 4, we introduce perturbative regularisations for Weyl fermions coupled to vector potentials and verify that the addition of a free Weyl fermions of opposite handedness allows us to define a functional integral for the system, while preserving the Weyl fermion's chirality. Section 5 is devoted to an introduction to quantum Majorana fermions. In Section 6, we recalculate consistent and covariant gauge anomalies for Weyl and Dirac fermions, by means of the Feynman diagram technique. In particular, in Section 7, we do the same calculation in a vector-axial background, and in Section 8 we apply these results to the case of Majorana fermions. In Section 9, we compute also the trace anomalies of Weyl fermions due to the presence of background gauge potentials and show that they are rigidly related to the previously calculated gauge anomalies. Section 10 is devoted to a summary of the results. Three Appendices contain auxiliary material.

Historical references for this review are [1–16].

*Notation*

We use a metric $g_{\mu\nu}$ with mostly $(-)$ signature. The gamma matrices satisfy $\{\gamma^\mu, \gamma^\nu\} = 2g^{\mu\nu}$ and

$$\gamma_\mu^\dagger = \gamma_0 \gamma_\mu \gamma_0.$$

At times we use also the $\alpha$ matrices, defined by $\alpha_\mu = \gamma_0 \gamma_\mu$. The generators of the Lorentz group are $\Sigma_{\mu\nu} = \frac{1}{4}[\gamma_\mu, \gamma_\nu]$. The charge conjugation operator $C$ is defined to satisfy

$$\gamma_\mu^T = -C^{-1}\gamma_\mu C, \qquad CC^* = -1, \qquad CC^\dagger = 1. \tag{1}$$

For example, $C = C^\dagger = C^{-1} = \gamma^0 \gamma^2 = \alpha^2$ does satisfy all the above requirements, but it holds only in some $\gamma$-matrix representations, such as the Dirac and Weyl ones, not in the Majorana. The chiral matrix $\gamma_5 = i\gamma^0 \gamma^1 \gamma^2 \gamma^3$ has the properties

$$\gamma_5^\dagger = \gamma_5, \qquad (\gamma_5)^2 = 1, \qquad C^{-1}\gamma_5 C = \gamma_5^T.$$

## 2. Dirac, Majorana and Weyl Fermions in 4D

Let us start from a few basic definitions and properties of spinors on a 4D Minkowski space [2]. A 4-component Dirac fermion $\psi$ under a Lorentz transformation transforms as

$$\psi(x) \to \psi'(x') = \exp\left[-\frac{1}{2}\lambda^{\mu\nu}\Sigma_{\mu\nu}\right]\psi(x),\tag{2}$$

for $x'^{\mu} = \Lambda^{\mu}{}_{\nu}x^{\nu}$. Here $\lambda^{\mu\nu} + \lambda^{\nu\mu} = 0$ are six real canonical coordinates for the Lorentz group, $\Sigma_{\mu\nu}$ are the generators in the 4D reducible representation of Dirac bispinors, while $\Lambda^{\mu}{}_{\nu}$ are the Lorentz matrices in the irreducible vector representation $D(\frac{1}{2}, \frac{1}{2})$. The invariant kinetic Lagrangian for a free Dirac field is

$$i\bar{\psi}\gamma^{\mu}\partial_{\mu}\psi.\tag{3}$$

where $\bar{\psi} = \psi^{\dagger}\gamma_0$.

A Dirac fermion admits a Lorentz invariant mass term $m\bar{\psi}\psi$.

A Dirac bispinor can be seen as the direct sum of two Weyl spinors

$$\psi_L = P_L\psi, \qquad \psi_R = P_R\psi, \qquad \text{where} \qquad P_L = \frac{1-\gamma_5}{2}, \qquad P_R = \frac{1+\gamma_5}{2}$$

with opposite chiralities

$$\gamma_5\psi_L = -\psi_L, \qquad \gamma_5\psi_R = \psi_R.$$

A left-handed Weyl fermion admits a Lagrangian kinetic term

$$i(\psi_L, \gamma\cdot\partial\psi_L) = i\overline{\psi}_L\gamma^{\mu}\partial_{\mu}\psi_L\tag{4}$$

but not a mass term, because $(\psi_L, \psi_L) = 0$, since $\gamma_5\gamma^0 + \gamma^0\gamma_5 = 0$. A Weyl fermion is massless and this property is protected by the chirality conservation.

For Majorana fermions we need the notion of *Lorentz covariant conjugate* spinor, $\hat{\psi}$:

$$\hat{\psi} = \gamma_0 C\psi^*.\tag{5}$$

It is not hard to show that if $\psi$ transforms like (2), then

$$\hat{\psi}(x) \to \hat{\psi}'(x') = \exp\left[-\frac{1}{2}\lambda^{\mu\nu}\Sigma_{\mu\nu}\right]\hat{\psi}(x).\tag{6}$$

Therefore one can impose on $\psi$ the condition

$$\psi = \hat{\psi}\tag{7}$$

because both sides transform in the same way. By definition, a spinor satisfying (7) is a Majorana spinor. It admits both kinetic and mass term.

It is a renowned fact that the group theoretical approach [1] to Atiyah Theory is one of the most solid and firm pillars in modern Physics. To this concern, the contributions by Eugene Paul Wigner were of invaluable importance [3]. In terms of Lorentz group representations we can say the following. $\gamma_5$ commutes with Lorentz transformations $\exp\left[-\frac{1}{2}\lambda^{\mu\nu}\Sigma_{\mu\nu}\right]$. So do $P_L$ and $P_R$. This means that the Dirac

---

[2]　This and the following section are mostly based on [17,18].

representation is reducible. Multiplying the spinors by $P_L$ and $P_R$ selects irreducible representations, the Weyl ones. To state it more precisely, Weyl representations are irreducible representations of the group $SL(2, C)$, which is the covering group of the proper ortochronous Lorentz group. They are usually denoted $(\frac{1}{2}, 0)$ and $(0, \frac{1}{2})$ in the $SU(2) \times SU(2)$ notation of the $SL(2, C)$ irreps. As we have seen in (6), Lorentz transformations commute also with the charge conjugation operation

$$\mathcal{C}\psi\mathcal{C}^{-1} = \eta_C \gamma_0 C \psi^* \tag{8}$$

where $\eta_C$ is a phase which, for simplicity, in the sequel we set equal to 1. This also implies that Dirac spinors are reducible and suggests another possible reduction: by imposing (7) we single out another irreducible representation, the Majorana one. The Majorana representation is the minimal irreducible representation of a (one out of eight) covering of the complete Lorentz group [5,11]. It is evident, and well-known, that Majorana and Weyl representations in 4D are incompatible.

Let us consider next the charge conjugation and parity and recall the relevant properties of a Weyl fermion. We have

$$\mathcal{C}\psi_L\mathcal{C}^{-1} = P_L\mathcal{C}\psi\mathcal{C}^{-1} = P_L\hat{\psi} = \hat{\psi}_L. \tag{9}$$

The charge conjugate of a Majorana field is, by definition, itself. While the action of a Majorana field is invariant under charge conjugation, for a Weyl fermion we have

$$\mathcal{C} \left( \int i\overline{\psi_L}\gamma^\mu\partial_\mu\psi_L \right) \mathcal{C}^{-1} = \int i\overline{\hat{\psi}_L}\gamma^\mu\partial_\mu\hat{\psi}_L = \int i\overline{\psi_R}\gamma^\mu\partial_\mu\psi_R. \tag{10}$$

i.e., a Weyl fermion is, so to say, maximally non-invariant.

The parity operation is defined by

$$\mathcal{P}\psi_L(t, \vec{x})\mathcal{P}^{-1} = \eta_P\gamma_0\psi_R(t, -\vec{x}) \tag{11}$$

where $\eta_P$ is a phase, which in the sequel we set to 1. In terms of the action we have

$$\mathcal{P} \left( \int \overline{\psi}_L\gamma^\mu\partial_\mu\psi_L \right) \mathcal{P}^{-1} = \int \overline{\psi}_R\gamma^\mu\partial_\mu\psi_R, \tag{12}$$

For a Majorana fermion the action is invariant under parity.

This also suggests a useful representation for a Majorana fermion. Let $\psi_R = P_R\psi$ be a generic Weyl fermion. We have $P_R\psi_R = \psi_R$ and it is easy to prove that $P_L\widehat{\psi_R} = \widehat{\psi_R}$, i.e., $\widehat{\psi_R}$ is left-handed. Therefore the sum $\psi_M = \psi_R + \widehat{\psi_R}$ is a Majorana fermion because it satisfies (7). Any Majorana fermion can be represented in this way. This representation is instrumental in the calculus of anomalies, see below.

Considering next CP, from the above it follows that the action of a Majorana fermion is obviously invariant under it. On the other hand, for a Weyl fermion we have

$$\mathcal{C}\mathcal{P}\psi_L(t, \vec{x})(\mathcal{C}\mathcal{P})^{-1} = \gamma_0 P_R\hat{\psi}(t, -\vec{x}) = \gamma_0\hat{\psi}_R(t, -\vec{x}). \tag{13}$$

Applying, now, CP to the action for a Weyl fermion, one gets

$$\mathcal{C}\mathcal{P} \left( \int i\overline{\psi_L}\gamma^\mu\partial_\mu\psi_L \right) (\mathcal{C}\mathcal{P})^{-1} = \int i\overline{\hat{\psi}_R}(t, -\vec{x})\gamma^{\mu\dagger}\partial_\mu\hat{\psi}_R(t, -\vec{x}) = \int i\overline{\hat{\psi}_R}(t, \vec{x})\gamma^\mu\partial_\mu\hat{\psi}_R(t, \vec{x}). \tag{14}$$

One can prove as well that

$$\int i\overline{\widehat{\psi}_R}(t, \vec{x})\gamma^\mu\partial_\mu\hat{\psi}_R(t, \vec{x}) = \int i\overline{\psi_L}(x)\gamma^\mu\partial_\mu\psi_L(x). \tag{15}$$

Therefore the action for a Weyl fermion is CP invariant. It is also, separately, T invariant, and, so, CPT invariant. The transformation properties of the Weyl and Majorana spinor fields are summarized in the following table:

$$
\begin{array}{lll}
 & \text{Majorana} & \text{Weyl} \\
\text{P}: & \mathcal{P}\psi(t,\vec{x})\mathcal{P}^{-1} = \gamma_0\psi(t,-\vec{x}) & \mathcal{P}\psi_L(t,\vec{x})\mathcal{P}^{-1} = \gamma_0\psi_R(t,-\vec{x}) \\
\text{C}: & \mathcal{C}\psi\mathcal{C}^{-1} = \gamma_0 C\psi^* = \psi & \mathcal{C}\psi_L\mathcal{C}^{-1} = P_L\hat{\psi} = \hat{\psi}_L \\
\text{CP}: & \mathcal{CP}\psi(t,\vec{x})(\mathcal{CP})^{-1} = \psi(t,-\vec{x}) & \mathcal{CP}\psi_L(t,\vec{x})(\mathcal{CP})^{-1} = \gamma_0\hat{\psi}_R(t,-\vec{x})
\end{array}
\tag{16}
$$

The quantum interpretation of the field $\psi_L$ starts from the plane wave expansion

$$
\psi_L(x) = \int dp \left( a(p)u_-(p)e^{-ipx} + b^\dagger(p)v_+(p)e^{ipx} \right)
\tag{17}
$$

where $u_-, v_+$ are fixed and independent left-handed spinors (there are only two of them). Such spin states are explicitly constructed in Appendix A. We interpret (17) as follows: $b^\dagger(p)$ creates a left-handed particle while $a(p)$ destroys a left-handed particle with negative helicity (because of the opposite momentum). However, Equations (13) and (14) force us to identify the latter with a right-handed antiparticle: C maps particles to antiparticles, while P invert helicities, so CP maps left-handed particles to right-handed antiparticles. One need not stress that in this game right-handed particles or left-handed antiparticles are absent.

**Remark 1.** *Let us comment on a few deceptive possibilities for a mass term for a Weyl fermion. A mass term $\bar{\psi}\psi$ for a Dirac spinor can also be rewritten by projecting the latter into its chiral components* [3]

$$
\bar{\psi}\psi = \overline{\psi_L}\psi_R + \overline{\psi_R}\psi_L.
\tag{18}
$$

*If $\psi$ is a Majorana spinor, $\psi = \hat{\psi}$, this can be rewritten as*

$$
\bar{\psi}\hat{\psi} = \overline{\psi_L}\hat{\psi}_R + \overline{\hat{\psi}_R}\psi_L,
\tag{19}
$$

*which is, by construction, well defined and Lorentz invariant. Now, by means of the Lorentz covariant conjugate, we can rewrite (19) as*

$$
(\psi_L)^T C^{-1}\psi_L + \psi_L^\dagger C(\psi_L)^*,
\tag{20}
$$

*which is expressed only in terms of $\psi_L$, although contains both chiralities. (20) may induce the impression that there exists a mass term also for Weyl fermions. This is not so. If we add this term to the kinetic term (4), the ensuing equation of motion is not Lorentz covariant: the kinetic and mass term in it belong to two different representations. To be more explicit, a massive Dirac equation of motion for a Weyl fermion should be*

$$
i\gamma^\mu\partial_\mu\psi_L - m\psi_L = 0,
\tag{21}
$$

*but it breaks Lorentz covariance because the first piece transforms according to the $(0,\frac{1}{2})$ representation, while the second according to $(\frac{1}{2},0)$, and is not Lagrangian* [4]. *The reason is, of course, that (20) is not expressible in*

---

[3]　Here and in the sequel we use $\bar{\psi}$ and $\overline{\psi}$, $\hat{\psi}$ and $\widehat{\psi}$ in a precise sense, which is hopefully unambiguous. For instance $\hat{\psi} = \psi^\dagger\gamma_0$, $\overline{\psi_R} = (\psi_R)^\dagger\gamma_0$, $\widehat{\psi_R} = \gamma_0 C(\psi_R)^*$, $\hat{\psi}_R = P_R\hat{\psi}$ and so on.

[4]　Instead of the second term in the LHS of (21) one could use $mC\overline{\psi_L}^T$, which has the right Lorentz properties, but the corresponding Lagrangian term would not be self-adjoint and one would be forced to introduce the adjoint term and end up again with (20). This implies, in particular, that there does not exist such a thing as a "massive Weyl propagator", that is a massive propagator involving only one chirality.

*the same canonical form as (4). This structure is clearly visible in the four component formalism used so far, but much less recognizable in the two-component formalism.*

*Weyl Fermions and Massless Majorana Fermions*

That a massive Majorana fermion and a Weyl fermion are different objects is uncontroversial. The question whether a massless Majorana fermion is or is not the same as a Weyl fermion at both the classical level and the quantum level is, instead, not always clear in the literature. Let us consider the simplest case in which there is no quantum number appended to the fermion. To start let us recall the obvious differences between the two. The first, and most obvious, has already been mentioned: they belong to two different irreducible representations of the Lorentz group (in 4D there cannot exist a spinor that is simultaneously Majorana and Weyl, like in 2D and 10D). Another important difference is that the helicity for a Weyl fermion is well defined and corresponds to its chirality, while for a Majorana fermion chirality is undefined, so that the relation with its helicity is also undefined. Then, a parity operation maps the Majorana action into itself, while it maps the Weyl action (4) into the same action for the opposite chirality. The same holds for the charge conjugation operator.　Why they are sometimes considered the same object may be due to the fact that we can establish a one-to-one correspondence between the components of a Weyl spinor and those of a Majorana spinor in such a way that the Lagrangian, in two-component notation, looks the same, see, for instance [16]. If, in the chiral representation, we write $\psi_L$ as $\begin{pmatrix} \omega \\ 0 \end{pmatrix}$, where $\omega$ is a two component spinor, then (4) above becomes

$$i\omega^\dagger \bar{\sigma}^\mu \partial_\mu \omega \qquad\qquad (22)$$

which has the same form (up to an overall factor) as a massless Majorana action (see Section 5 below and Equation (49)). The action is not everything in a theory, it must be accompanied by a set of specifications. Even though numerically the actions coincide, the way they respond to a variation of the Weyl and Majorana fields is different. One leads to the Weyl equation of motion, the other to the Majorana equation of motion. The delicate issue is precisely this: when we take the variation of an action with respect to a field in order to extract the equations of motion, we must make sure that the variation respects the symmetries and the properties that are expected in the equations of motion [5]. Specifically in the present case, if we wish the equation of motion to preserve chirality we must use variations that preserve chirality, i.e., variations that are eigenfunctions of $\gamma_5$. If instead we wish the equation of motion to transform in the Majorana representation we have to use variations that transform suitably, i.e., variations that are eigenfunctions of the charge conjugation operator. If we do so, we obtain two different results which are irreducible to each other no matter which action we use.

There is no room for confusing massless Majorana spinors with chiral Weyl spinors. A classical Majorana spinor is a self-conjugated bispinor, that can always be chosen to be real and always contains both chiralities in terms of four real independent component functions. It describes neutral spin 1/2 objects—not yet detected in Nature—and consequently there is no phase transformation (U(1) continuous symmetry) involving self-conjugated Majorana spinors, independently of the presence or not of a mass term. Hence, e.g., its particle states do not admit antiparticles of opposite charge, simply because charge does not exist at all for charge self-conjugated spinors (actually, this was the surprising discovery of Ettore Majorana, after the appearance of the Dirac equation and the positron detection). The general solution of the wave field equations for a free Majorana spinor always entails the presence of two polarization states with opposite helicity. On the contrary, it is well known that a chiral Weyl spinor, describing massless neutrinos in the Standard Model, admits only one polarization or helicity

---

[5]　For instance, in gravity theories, the metric variation $\delta g_{\mu\nu}$ is generic while not ceasing to be a symmetric tensor.

state, it always involves antiparticles of opposite helicity and it always carries a conserved internal quantum number such as the lepton number, which is opposite for particles and antiparticles.

Finally, and most important, in the quantum theory a crucial role is played by the functional measure, which is different for Weyl and Majorana fermions. We will shortly come back to this point. Before that, it is useful to clarify an issue concerning the just mentioned U(1) continuous symmetry of Weyl fermions. The latter is sometime confused with the axial $\mathbb{R}$ symmetry of Majorana fermions and assumed to justify the identification of Weyl and massless Majorana fermions. To start with, let us consider a free massless Dirac fermion $\psi$. Its free action is clearly invariant under the transformation $\delta\psi = i(\alpha + \gamma_5\beta)\psi$, where $\alpha$ and $\beta$ are real numbers. This symmetry can be gauged by minimally coupling $\psi$ to a vector potential $V_\mu$ and an axial potential $A_\mu$, in the combination $V_\mu + \gamma_5 A_\mu$, so that $\alpha$ and $\beta$ become arbitrary real functions. For convenience, let us choose the Majorana representation for gamma matrices, so that all of them, including $\gamma_5$, are imaginary. If we now impose $\psi$ to be a Majorana fermion, its four component can be chosen to be real and only the symmetry parametrized by $\beta$ makes sense in the action (let us call it $\beta$ symmetry). If instead we impose $\psi$ to be Weyl, say $\psi = \psi_L$, then , since $\gamma_5\psi_L = \psi_L$, the symmetry transformation will be $\delta\psi_L = i(\alpha - \beta)\psi_L$.

We believe this may be the origin of the confusion, because it looks like we can merge the two parameters $\alpha$ and $\beta$ into a single parameter identified with the $\beta$ of the Majorana axial $\beta$ symmetry. However this is not correct because for a right handed Weyl fermion the symmetry transformation is $\delta\psi_R = i(\alpha + \beta)\psi_R$. Forgetting $\beta$, the Majorana fermion does not transform. Forgetting $\alpha$, both Weyl and Majorana fermions transform, but the Weyl fermions transform with opposite signs for opposite chiralities. This distinction will become crucial in the computation of anomalies (see below).

## 3. Functional Integral for Dirac, Weyl and Majorana Fermions

In quantum field theory there is one more reason to distinguish between massless Weyl and Majorana fermions: their functional integration measure is formally and substantially different. Although the action in the two-component formalism may take the same form (22) for both, the change of integration variable from $\psi_L$ to $\omega$ is not an innocent field redefinition because the functional integration measure changes. The purpose of this section is to illustrate this issue. To start with, let us clarify that speaking about functional integral measure is a colourful but not rigorous parlance. The real issue here is the definition of the functional determinant for a Dirac-type matrix-valued differential operator.

Let us start with some notations and basic facts. We denote by $\slashed{D}$ the standard Dirac operator: namely, the massless matrix-valued differential operator applied in general to Dirac spinors on the 4D curved space with Minkowski signature $(+, -, -, -)$

$$\slashed{D} = i\left(\slashed{\partial} + \slashed{V}\right) \tag{23}$$

where $V_\mu$ is any anti-Hermitean vector potential, including a spin connection in the presence of a non-trivial background metric. We use here the four component formalism for fermions. The functional integral, i.e., the effective action for a quantum Dirac spinor in the presence of a classical background potential

$$\mathcal{Z}[V] = \int \mathcal{D}\psi \mathcal{D}\bar{\psi}\, e^{i \int d^4x \sqrt{g}\, \bar{\psi}\slashed{D}\psi} \tag{24}$$

is formally understood as the determinant of $\slashed{D}$ : $\det\left(\slashed{D}\right) = \prod_i^\infty \lambda_i$. From a concrete point of view, the latter can be operatively defined in two alternative ways: either in perturbation theory, i.e., as the sum of an infinite number of 1-loop Feynman diagrams, some of which contain UV divergences by naïve power counting, or by a non-perturbative approach, i.e., as the suitably regularised infinite product of the eigenvalues of $\slashed{D}$ by means of the analytic continuation tool. It is worthwhile to remark that, on the one hand, the perturbative approach requires some UV regulator and renormalisation prescription, in

order to give a meaning to a finite number of UV divergent 1-loop diagrams by naïve power counting. On the the other hand, in the non-perturbative framework the complex power construction and the analytic continuation tool, if available, provide by themselves the whole necessary setting up to define the infinite product of the eigenvalues of a normal operator, without need of any further regulator.

In many practical calculations one has to take variations of (24) with respect to $V$. In turn, any such variation requires the existence of an inverse of the kinetic operator, as follows from the abstract formula for the determinant of an operator $A$

$$\delta \det A = \det A \operatorname{tr}\left(A^{-1}\delta A\right).$$

It turns out that an inverse of $\slashed{D}$ does exist and, if full causality is required in forwards and backwards time evolution on e.g., Minkowski space, it is the Feynman propagator or Schwinger distribution $\slashed{S}$, which is unique and characterized by the well-known Feynman prescription, in such a manner that

$$\slashed{D}_x \slashed{S}(x-y) = \delta(x-y), \qquad \slashed{D}\slashed{S} = 1. \tag{25}$$

The latter is a shortcut operator notation, which we are often going to use in the sequel [6].

For instance, the scheme to extract the trace of the stress-energy tensor from the functional integral is well-known. It is its response under a Weyl (or even a scale) transform $\delta_\omega g_{\mu\nu} = 2\omega g_{\mu\nu}$ :

$$\delta_\omega \log \mathcal{Z} = \int d^4 x\, \omega(x)\, g_{\mu\nu}(x)\, \langle\, T^{\mu\nu}(x)\,\rangle \tag{26}$$

where $g_{\mu\nu}(x)\langle T^{\mu\nu}(x)\rangle$ is the quantum trace of the energy-momentum tensor. Analogously, the divergence of the vector current $j_\mu = \bar{\psi}\gamma_\mu\psi$ is the response of $\log \mathcal{Z}$ under the Abelian gauge transformation $\delta_\lambda V_\mu = \partial_\mu \lambda$:

$$\delta_\lambda \log \mathcal{Z} = -i \int d^4 x\, \lambda(x)\partial_\mu \langle j^\mu(x)\rangle \tag{27}$$

and so on. These quantities can be calculated in various ways with perturbative or non-perturbative methods. The most frequently used ones are the Feynman diagram technique and the so-called analytic functional method, respectively. The latter denomination actually includes a collection of approaches, ranging from the Schwinger's proper-time method [6] to the heat kernel method [13], the Seeley-DeWitt [7,8] and the zeta-function regularisation [10]. The central tool in these approaches is the (full) kinetic operator of the fermion action (or the square thereof), and its inverse, the full fermion propagator. All these methods yield well-known results with no disagreement among them.

On the contrary, when one comes to Weyl fermions things drastically change. The classical action on the 4D Minkowski space for a left-handed Weyl fermion reads

$$S_L = \int d^4 x\, \bar{\psi}_L \slashed{D} \psi_L. \tag{28}$$

The Dirac operator, acting on left-handed spinors maps them to right-handed ones. Hence, the Sturm–Liouville or eigenvalue problem itself is not well posed, so that the Weyl determinant cannot even be defined. This is reflected in the fact that the inverse of $\slashed{D}_L = \slashed{D}P_L = P_R\slashed{D}$ does not exist, since it is the product of an invertible operator times a projector. As a consequence the full propagator

---

[6]    For simplicity we understand factors of $\sqrt{g}$, which should be there, see [7], but are inessential in this discussion.

of a Weyl fermion does not exist in this naïve form (this problem can be circumvented in a more sophisticated approach, see below) [7].

The lack of an inverse for the chiral Dirac–Weyl kinetic term has drastic consequences even at the free non-interacting level. For instance, the evaluation of the functional integral (i.e., formally integrating out the spinor fields) involves the inverse of the kinetic operator: thus, it is clear that the corresponding formulas for the chiral Weyl quantum theory cannot exist at all, so that no Weyl effective action can be actually defined in this way even in the free non-interacting case. Let us add that considering the square of the kinetic operator, as it is often done in the literature, does not change this conclusion.

It may sound strange that the (naïve) full propagator for Weyl fermions does not exist, especially if one has in mind perturbation theory in Minkowski space. In that case, in order to construct Feynman diagrams, one uses the ordinary free Feynman propagator for Dirac fermions. The reason one can do so is because the information about chirality is preserved by the fermion–boson–fermion vertex, which contains the $P_L$ projector (the use of a free Dirac propagator is formally justified, because one can add a free right-handed fermion to allow the inversion of the kinetic operator, see below). On the contrary, the full (non-perturbative) propagator is supposed to contain the full chiral information, including the information contained in the vertex, i.e., the potential, as it will be explicitly checked here below. In this problem there is no simple shortcut such as pretending to replace the full Weyl propagator with the full Dirac propagator multiplied by a chiral projector, because this would destroy any information concerning the chirality.

The remedy for the Weyl fermion disaster is to use as kinetic operator

$$i\gamma^\mu \left( \partial_\mu + P_L V_\mu \right),\tag{30}$$

which is invertible and in accord with the above mentioned Feynman diagram approach. It corresponds to the intuition that the free right-handed fermions added to the left-handed theory in this way do not interfere with the conservation of chirality and do not alter the left-handed nature of the theory. It is important to explicitly check it. The next section is devoted to a close inspection of this problem and its solution.

## 4. Regularisations for Weyl Spinors

The classical Lagrange density for a Weyl (left) spinor in the four component formalism

$$\psi(x) = \chi_L(x) = \left( \begin{array}{c} \chi(x) \\ 0 \end{array} \right)$$

reads

$$\mathcal{K}(x) = \overline{\psi}(x)\, i\slashed{\partial}\, \psi(x) = \chi_L^\dagger(x)\, \alpha^\nu i\partial_\nu \chi_L(x).\tag{31}$$

It follows that the corresponding matrix valued Weyl differential operator

$$w_L \equiv \alpha^\nu i\partial_\nu P_L\tag{32}$$

---

[7]    It is incorrect to pretend that the propagator is $\slashed{S}_L = \slashed{S} P_R = P_L \slashed{S}$. First because such an inverse does not exist, second because, even formally,

$$\slashed{D}_L \slashed{S}_L = P_R, \qquad \text{and} \qquad \slashed{S}_L \overleftarrow{\slashed{D}}_L = P_L\tag{29}$$

The inverse of the Weyl kinetic operator is not the inverse of the Dirac operator multiplied by a chiral projector. Therefore the propagator for a Weyl fermion is not the Feynman propagator for a Dirac fermion multiplied by the same projector.

is singular and does not possess any rank-four inverse. After minimal coupling with a real massless vector field $A^\mu(x)$ we come to the classical Lagrangian

$$\mathcal{L} = \chi_L^\dagger \, \alpha^\nu i\partial_\nu \chi_L + g A^\nu \chi_L^\dagger \, \alpha_\nu \chi_L - \tfrac{1}{4} F^{\mu\nu} F_{\mu\nu} \tag{33}$$

where $F_{\mu\nu} = \partial_\mu A_\nu - \partial_\nu A_\mu$. It turns out that the classical action

$$S = \int \mathrm{d}^4 x \, \mathcal{L} \tag{34}$$

is invariant under the Poincaré group, as well as under the internal U(1) phase transformations $\chi_L(x) \mapsto e^{ig\theta}\chi_L(x)$. The action integral is invariant under the so called scale or dilatation transformations, viz.,

$$x'^\mu = e^{-\varrho} x^\mu \qquad \chi_L'(x) = e^{\frac{3}{2}\varrho}\chi_L(e^\varrho x) \qquad A'^\mu(x) = e^\varrho A^\mu(e^\varrho x)$$

with $\varrho \in \mathbb{R}$, as well as with respect to the local phase or gauge transformations

$$\chi_L'(x) = e^{ig\theta(x)}\chi_L(x) \qquad A_\nu'(x) = A_\nu(x) + \partial_\nu \theta(x)$$

which amounts to the ordinary U(1) phase transform in the limit of constant phase. It follows therefrom that there are twelve conserved charges in this model at the classical level and, in particular, owing to scale and gauge invariance, no mass term is allowed for both spinor and vector fields. The question naturally arises if those symmetries hold true after the transition to the quantum theory and, in particular, if they are protected against loop radiative corrections within the perturbative approach. Now, as explained above, in order to develop perturbation theory, one faces the problem of the lack of an inverse for both the Weyl and gauge fields, owing to chirality and gauge invariance. In order to solve it, it is expedient to add to the Lagrangian non-interacting terms, which are fully decoupled from any physical quantity. They break chirality and gauge invariance, albeit in a harmless way, just to allow us to define a Feynman propagator, or causal Green's functions, for both the Weyl and gauge quantum fields. The simplest choice, which preserves Poincaré and internal U(1) phase change symmetries, is provided by

$$\mathcal{L}' = \varphi_R^\dagger \, \alpha^\nu i\partial_\nu \varphi_R - \tfrac{1}{2}(\partial \cdot A)^2$$

where

$$\psi(x) = \varphi_R(x) = \begin{pmatrix} 0 \\ \varphi(x) \end{pmatrix}$$

is a left-chirality breaking right-handed Weyl spinor field. Notice *en passant* that the modified Lagrangian $\mathcal{L} + \mathcal{L}'$ exhibits a further U(1) internal symmetry under the so called chiral phase transformations

$$\psi'(x) = (\cos\theta + i\sin\theta \, \gamma_5)\psi(x) \qquad \psi(x) = \begin{pmatrix} \chi(x) \\ \varphi(x) \end{pmatrix}$$

so that the modified theory involves another conserved charge at the classical level. From the modified Lagrange density we get the Feynman propagators for the massless Dirac field $\psi(x)$, as well as for the massless vector field in the so called Feynman gauge: namely,

$$S(p) = \frac{i p\!\!\!/}{p^2 + i\varepsilon} \qquad D_{\mu\nu}(k) = \frac{-i\eta_{\mu\nu}}{k^2 + i\varepsilon} \tag{35}$$

and the vertex $ig\gamma^\nu P_L$, with $k + p - q = 0$, which involves a vector particle of momentum $k$ and a Weyl pair of particle and anti-particle of momenta $p$ and $q$, respectively, and of opposite helicity. [8]

Our purpose hereafter is to show that, notwithstanding the use of the non-chiral propagators (35), a mass in the Weyl kinetic term cannot arise as a consequence of quantum corrections. The lowest order 1-loop correction to the kinetic term $\not{k}P_L$ is provided by the Feynman rules in Minkowski space, in the following form

$$\Sigma_2(\not{k}) = -ig^2 \int \frac{\mathrm{d}^4\ell}{(2\pi)^4}\, \gamma^\mu \, D_{\mu\nu}(k-\ell)\, S(\ell)\, \gamma^\nu P_L. \tag{36}$$

A mass term in this context should be proportional to the identity matrix (in the spinor space).

By naïve power counting the above 1-loop integral turns out to be UV divergent. Hence, a regularisation procedure is mandatory to give a meaning and evaluate the radiative correction $\Sigma_2(\not{k})$ to the Weyl kinetic operator. Here in the sequel we shall examine in detail the dimensional, Pauli–Villars and UV cut-off regularisations.

### 4.1. Dimensional, PV and Cutoff Regularisations

In a $2\omega$-dimensional space-time, the dimensionally regularised radiative correction to the Weyl kinetic term takes the form

$$\mathtt{reg}\,\Sigma_2(\not{k}) = -ig^2\mu^{2\epsilon} \int \frac{\mathrm{d}^{2\omega}\ell}{(2\pi)^{2\omega}}\, D_{\mu\nu}(\ell)\, \gamma^\mu\, S(\ell+k)\, \gamma^\nu P_L \tag{37}$$

where $\epsilon = 2 - \omega > 0$ is the shift with respect to the physical space-time dimensions. Since the above expression is traceless and has the canonical engineering dimension of a mass in natural units, it is quite apparent that the latter cannot generate any mass term, which, as anticipated above, would be proportional to the unit matrix. Hence, mass is forbidden and it remains for us to evaluate

$$\mathtt{reg}\,\Sigma_2(\not{k}) \;\equiv\; f(k^2)\,\not{k}P_L \qquad \mathrm{tr}\,[\not{k}\,\mathtt{reg}\,\Sigma_2(\not{k})] = \tfrac{1}{2}\,2^\omega k^2 f(k^2) \tag{38}$$

$$\mathrm{tr}\,[\not{k}\,\mathtt{reg}\,\Sigma_2(\not{k})] \;=\; g^2\mu^{2\epsilon}\,(2\pi)^{-2\omega} \int \mathrm{d}^{2\omega}\ell\, \frac{(-i)\,\mathrm{tr}\,(\not{k}\gamma^\lambda\not{\ell}\gamma_\lambda P_L)}{[\,(\ell-k)^2 + i\varepsilon\,]\,(\ell^2 + i\varepsilon\,)}. \tag{39}$$

For $2^\omega \times 2^\omega$ $\gamma$-matrix traces in a $2\omega-$dimensional space-time with a Minkowski signature the following formulas are necessary

$$\mathrm{tr}\,(\gamma^\mu\gamma^\nu) \;=\; g^{\mu\nu}\,\mathrm{tr}\,\mathbb{I} = 2^\omega\,g^{\mu\nu} \tag{40}$$

$$2^{-\omega}\mathrm{tr}\,\left(\gamma^\kappa\gamma^\lambda\gamma^\mu\gamma^\nu\right) \;=\; g^{\kappa\lambda}\,g^{\mu\nu} - g^{\kappa\mu}\,g^{\lambda\nu} + g^{\kappa\nu}\,g^{\lambda\mu}. \tag{41}$$

Then we get $\mathrm{tr}\,(\not{k}\gamma^\lambda\not{\ell}\gamma_\lambda P_L) = 2^\omega(\epsilon-1)\,p\cdot\ell$ and thereby

$$k^2 f(k^2) = ig^2\mu^{2\epsilon}\,\frac{\epsilon-1}{(2\pi)^{2\omega}} \int \frac{2p\cdot\ell\,\mathrm{d}^{2\omega}\ell}{[\,(\ell-k)^2 + i\varepsilon\,]\,(\ell^2 + i\varepsilon\,)}. \tag{42}$$

Turning to the Feynman parametric representation we obtain

$$k^2 f(k^2) = ig^2\mu^{2\epsilon}\,\frac{\epsilon-1}{(2\pi)^{2\omega}} \int_0^1 \mathrm{d}x \int \frac{2p\cdot\ell\,\mathrm{d}^{2\omega}\ell}{[\,\ell^2 - 2x\,k{\cdot}\ell + xk^2 + i\varepsilon\,]^2}. \tag{43}$$

---

[8]　Customarily, the on-shell 1-particle states of a left Weyl spinor field are a left-handed particle with negative helicity $-\tfrac{1}{2}\hbar$ and a right-handed antiparticle of positive helicity $\tfrac{1}{2}\hbar$.

Completing the square in the denominator and after shifting the momentum $\ell' \equiv \ell - xp$, dropping the linear term in $\ell'$ in the numerator owing to symmetric integration, we have

$$f(k^2) = 2ig^2\mu^{2\epsilon}\frac{\epsilon-1}{(2\pi)^{2\omega}}\int_0^1 dx\,x\int\frac{d^{2\omega}\ell}{[\ell^2+x(1-x)k^2+i\varepsilon]^2}. \tag{44}$$

One can perform the Wick rotation and readily get the result

$$\begin{aligned}
f(k^2) &= -2g^2\mu^{2\epsilon}\frac{\epsilon-1}{(4\pi)^{\omega}}\int_0^1 dx\,x\int_0^\infty d\tau\,\tau^{\epsilon-1}e^{-\tau x(1-x)k_E^2}\\
&= 2\left(\frac{g}{4\pi}\right)^2[\Gamma(\epsilon)-\Gamma(1+\epsilon)]\left(-\frac{4\pi\mu^2}{k^2}\right)^\epsilon B(2-\epsilon,1-\epsilon).
\end{aligned} \tag{45}$$

Expansion around $\epsilon = 0$ yields

$$f(k^2) = \left(\frac{g}{4\pi}\right)^2\left[\frac{1}{\epsilon}+1+3\mathbf{C}+\ln\left(-\frac{4\pi\mu^2}{k^2}\right)\right]+\text{evanescent} \tag{46}$$

where $\mathbf{C}$ denotes the Euler–Mascheroni constant.

Similar results are obtained with the Pauli–Villars and cut-off regularisations. In the PV case the latter is simply implemented by the following replacement of the massless Dirac propagator

$$\text{reg}\,\Sigma_2(k) = -ig^2\int\frac{d^4\ell}{(2\pi)^4}\,\gamma^\mu D_{\mu\nu}(k-\ell)\sum_{s=0}^S C_s S(\ell,M_s)\,\gamma^\nu P_L, \tag{47}$$

where $M_0 = 0$, $C_0 = 1$ while $\{M_s \equiv \lambda_s M\,|\,\lambda_s \gg 1\,(s=1,2,\dots,S)\}$ is a collection of very large auxiliary masses. The constants $C_s$ are required to satisfy:

$$\sum_{s=1}^S C_s = -1 \qquad \sum_{s=1}^S C_s\lambda_s = 0$$

and the following identification with the divergent parameter is made

$$\frac{1}{\epsilon} = \sum_{s=1}^S C_s\ln\lambda_s.$$

The result for $f(k^2)$ is

$$f(k^2) = \left(\frac{g}{4\pi}\right)^2\left[\sum_{s=1}^S C_s\ln\lambda_s+\frac{1}{4}+\frac{1}{2}\ln\left(-\frac{M^2}{k^2}\right)\right]+\text{evanescent}. \tag{48}$$

The same calculation can be repeated with an UV cutoff $K$, see [18]. To sum up, we have verified that the 1-loop correction to the (left) Weyl spinor self-energy has the general form, which is universal, i.e., regularisation independent: namely,

$$\begin{aligned}
\text{reg}\,\Sigma_2(k) &\equiv f(k^2)\,k P_L\\
f(k^2) &:= \left(\frac{g}{4\pi}\right)^2\left[\frac{1}{\epsilon}+1+3\mathbf{C}+\ln\left(-\frac{4\pi\mu^2}{k^2}\right)\right] \qquad (\text{DR})\\
&:= \left(\frac{g}{4\pi}\right)^2\left[\sum_{s=1}^S C_s\ln\lambda_s+\frac{1}{4}+\frac{1}{2}\ln\left(-\frac{M^2}{k^2}\right)\right] \qquad (\text{PV})\\
&:= \left(\frac{g}{4\pi}\right)^2\ln\left[-\frac{(4K)^2}{k^2}\right] \qquad (\text{CUT}-\text{OFF})
\end{aligned}$$

Remarks

1.  In the present model of a left Weyl spinor minimally coupled to a gauge vector potential, no mass term can be generated by the radiative corrections in any regularisation scheme. The left-handed part of the classical kinetic term does renormalise, while its right-handed part does not undergo any radiative correction and keeps on being free. The latter has to be necessarily introduced in order to define a Feynman propagator for the massless spinor field, much like the gauge fixing term is introduced in order to invert the kinetic term of the gauge potential. The (one loop) renormalised Lagrangian for a Weyl fermion minimally coupled to a gauge vector potential has the universal—i.e., regularisation independent—form

$$
\begin{aligned}
\mathcal{L}_{\text{ren}} \;=\;& \chi_L^\dagger \, \alpha^\nu i\partial_\nu \chi_L + g A^\nu \, \chi_L^\dagger \, \alpha_\nu \chi_L - \tfrac{1}{4} F^{\mu\nu} F_{\mu\nu} \\
+\;& \varphi_R^\dagger \, \alpha^\nu i\partial_\nu \varphi_R - \tfrac{1}{2}(\partial \cdot A)^2 - (Z_3 - 1)\tfrac{1}{4} F^{\mu\nu} F_{\mu\nu} \\
+\;& (Z_2 - 1)\chi_L^\dagger \, \alpha^\nu i\partial_\nu \chi_L + (Z_1 - 1)g A^\nu \, \chi_L^\dagger \, \alpha_\nu \chi_L
\end{aligned}
$$

$$
\begin{aligned}
(Z_2 - 1)_{1-\text{loop}} \;=\;& -\left(\frac{g}{4\pi}\right)^2 \left[\frac{1}{\epsilon} + F_2(\epsilon, k^2/\mu^2)\right] \\
=\;& -\left(\frac{g}{4\pi}\right)^2 \left[\sum_{s=1}^{S} C_s \ln \lambda_s + \widetilde{F}_2(\lambda_s, k^2/M^2)\right] \\
=\;& -\left(\frac{g}{4\pi}\right)^2 \left\{\ln\left[-\frac{(4K)^2}{k^2}\right] + \widehat{F}_2(K^2/k^2)\right\}
\end{aligned}
$$

    where the customary notations have been employed. Notice that the arbitrary finite parts $F_2, \widetilde{F}_2, \widehat{F}_2$ of the counterterms are analytic for $\epsilon \to 0$ and $\lambda_s, K \to \infty$, respectively, and have to be univocally fixed by the renormalisation prescription, as usual.

2.  The interaction definitely preserves left chirality and scale invariance of the counterterms in the transition from the classical to the (perturbative) quantum theory: no mass coupling between the left-handed (interacting) Weyl spinor $\chi_L$ and right-handed (free) Weyl spinor $\varphi_R$ can be generated by radiative loop corrections.

3.  While the cut-off and dimensional regularised theory does admit a local formulation in $D = 4$ or $D = 2\omega$ space-time dimensions, there is no such local formulation for the Pauli–Villars regularisation. The reason is that the PV spinor propagator

$$
\sum_{s=0}^{S} C_s \, S(\ell, M_s)
$$

    where $M_0 = 0$, $C_0 = 1$ while $\{\, M_s \equiv \lambda_s M \,|\, \lambda_s \gg 1 \,(s = 1, 2, \ldots, S)\,\}$, cannot be the inverse of any local differential operator of the Calderon–Zygmund type. Hence, there is no local action involving a bilinear spinor term that can produce, after a suitable inversion, the Pauli–Villars regularised spinor propagator. Although the Seeley–Schwinger–DeWitt method is not the main concern of this paper, there are no doubts that the Pauli–Villars regularisation cannot be applied to the construction of a regularised full kinetic operator for the Seeley–Schwinger–DeWitt method, nor, of course, to its inverse.

## 5. Majorana Massless Quantum Field

One can write a relativistic invariant field equation for a massless 2-component spinor field: to this aim, let us start from a left Weyl spinor $\psi_L \in D(\tfrac{1}{2}, 0)$ that transforms according to the $SL(2, \mathbb{C})$ matrix $\Lambda_L$. Call such a 2-component spinor field $\chi_a(x)$ $(a = 1, 2)$. Let us consider the Weyl spinor wave field as a classical anti-commuting field. A Majorana classical spinor field is a self-conjugated

bispinor, that can be constructed, for example, out of the left-handed spinor $\chi_a(x)$ ($a = 1, 2$) as follows: namely,

$$\chi_M(x) = \begin{pmatrix} \chi(x) \\ -\sigma_2 \chi^*(x) \end{pmatrix} = \chi_M^c(x)$$

the charge conjugation rule for any classical bispinors $\psi$ being defined by the general relationship

$$\psi^c(x) = e^{i\theta} \gamma^2 \psi^*(x) \qquad (0 \le \theta < 2\pi)$$

which is a discrete internal—i.e., space-time point independent—symmetry transformation. Here below we shall suitably choose $\theta = 0$. The Majorana bispinor has a right-handed lower Weyl spinor component $-\sigma_2 \chi^* \in D(0, \frac{1}{2})$, albeit functional dependent, due to the charge self-conjugation constraint, in such a manner that $\chi_M$ possesses both chiralities and polarizations, at variance with its left-handed Weyl building spinor $\chi(x)$. There is another kind of self-conjugated Majorana bispinor, which can be set-up out of a right-handed Weyl building spinor $\varphi \in D(0, \frac{1}{2})$.

From the Majorana self-conjugated bispinors, one can readily construct the most general Poincaré invariant and power counting renormalisable Lagrangian. For instance, by starting from the bispinor $\chi_M(x)$ we have

$$\mathcal{L}_M = \tfrac{1}{4} \overline{\chi}_M(x) \gamma^\mu i \overset{\leftrightarrow}{\partial}_\mu \chi_M^c(x)$$

where $\alpha^\nu = \gamma_0 \gamma^\nu$, while the employed notation reminds us that the upper and lower components of a Majorana bispinor can never be treated as functionally independent, even formally, due to the presence of the self-conjugation constraint. It follows that the massless Majorana action integral

$$\int \mathrm{d}^4 x \, \chi_M^\dagger(x) \tfrac{1}{4} \alpha^\mu \gamma^2 i \overset{\leftrightarrow}{\partial}_\mu \chi_M^*(x)$$

is not invariant under the overall phase transformation $\chi_M'(x) = e^{i\theta} \chi_M(x)$ of the Majorana bispinor. Hence it turns out that, as it will be further endorsed after the transition to the Majorana representation of the Dirac matrices, there is no invariant scalar charge for a Majorana spinor, which is a genuinely neutral spin $\frac{1}{2}$ field. As it will be clarified in the sequel, there is a relic continuous U(1) symmetry only for Majorana massless spinors, which drives to the existence of a conserved *pseudo-scalar charge*, the meaning of which will be better focused further on.

The massless Majorana Lagrangian in the 2-component formalism reads

$$\widehat{\mathcal{L}}_M = \tfrac{1}{2} \chi^\dagger(x) \sigma^\mu i\partial_\mu \chi(x) + \text{c.c.} \tag{49}$$

so that the Euler-Lagrange field equation may be written in the equivalent forms

$$i\sigma^\mu \partial_\mu \chi(x) = 0 \qquad i\bar{\sigma}^\mu \sigma_2 \partial_\mu \chi^*(x) = 0 \tag{50}$$

which are nothing but the pair of the Weyl wave equations for both a left-handed Weyl spinor $\chi(x)$ and a right-handed Weyl spinor $\sigma_2 \chi^*(x)$. This means, of course, that a massless Majorana spinor field always involves a pair of Weyl spinor fields—albeit functional dependent due to the self-conjugation constraint—with opposite chirality. As a further consequence we find that

$$\bar{\sigma}^\nu \sigma^\mu \partial_\nu \partial_\mu \chi(x) = \Box \chi(x) = 0$$

which means that the left-handed spinor $\chi \in D(\frac{1}{2}, 0)$, the building block of the self-conjugated Majorana bispinor, is actually solution of the d'Alembert wave equation. It is easy to check that the pair of Equation (50) is equivalent to the single bispinor equation

$$\alpha^\nu i\partial_\nu \chi_M(x) = 0 \tag{51}$$

where use has been made of the Dirac notation $\beta = \gamma_0$, while the Majorana Lagrangian can be recast in a further 4-component form

$$\mathcal{L}_M = \tfrac{1}{4}\, \chi_M^\dagger(x)\, \alpha^\mu\, i\, \overset{\leftrightarrow}{\partial}_\mu \chi_M^c(x) \tag{52}$$

Notice that the Majorana self-conjugated bispinor transforms under the Poincaré group as

$$\chi_M'(x') = \Lambda_{\frac{1}{2}}\, \chi_M^c(x) \qquad \Lambda_{\frac{1}{2}} = \left( \begin{array}{cc} \Lambda_L & 0 \\ 0 & \Lambda_R \end{array} \right) \tag{53}$$

with $x' = \Lambda(x + a)$, $\Lambda$ being the Lorentz matrices in the vector representation and $a^\mu$ a constant space-time translation.

It turns out that, by definition, the Majorana bispinor $\chi_M(x) = \chi_M^c(x)$ must fulfil the self-conjugation constraint, which linearly relates the lower spinor component to the complex conjugate of the upper spinor component. Then, a representation must exist which makes the Majorana bispinor real, in such a manner that the previously introduced pair of complex variables $\chi_a \in \mathbb{C}$ $(a = 1,2)$ could be replaced by the four real variables $\psi_{M,\alpha} \in \mathbb{R}$ $(\alpha = 1,2,3,4)$. To obtain this real representation, we note that

$$\chi_M = \left( \begin{array}{c} \chi \\ -\sigma_2\, \chi^* \end{array} \right) \qquad \chi_M^* = \left( \begin{array}{cc} 0 & -\sigma_2 \\ \sigma_2 & 0 \end{array} \right) \chi_M = -\gamma^2 \chi_M$$

A transformation to *real* bispinor fields $\psi_M = \psi_M^*$ can be made by writing

$$\chi_M = S\, \psi_M \qquad \chi_M^* = S^*\, \psi_M = S^* S^{-1}\, \chi_M$$

with

$$S = \frac{\sqrt{2}}{2} \left( \begin{array}{cccc} 1 & 0 & 0 & -i \\ 0 & 1 & i & 0 \\ 0 & i & 1 & 0 \\ -i & 0 & 0 & 1 \end{array} \right)$$

From the above relation $\psi_M = S^{-1}\chi_M = \psi_M^*$ one can immediately obtain the correspondence rule between the complex and real forms of the self-conjugated Majorana bispinor: namely,

$$\begin{array}{cc} \psi_{M1} = \sqrt{2}\,\Re\mathfrak{e}\,\chi_1 & \psi_{M2} = \sqrt{2}\,\Re\mathfrak{e}\,\chi_2 \\ \psi_{M3} = -\sqrt{2}\,\Im\mathfrak{m}\,\chi_2 & \psi_{M4} = \sqrt{2}\,\Im\mathfrak{m}\,\chi_1 \end{array}$$

Thus we can make use of the so called Majorana representation for the Clifford algebra which is given by the similarity transformation acting on the $\gamma$-matrices in the Weyl representation

$$\gamma_M^\mu \equiv S^\dagger \gamma^\mu S$$

which satisfy by direct inspection

$$\{\gamma_M^\mu, \gamma_M^\nu\} = 2\eta^{\mu\nu} \qquad \{\gamma_M^\nu, \gamma_M^5\} = 0$$

$$\gamma_M^0 = \beta_M^\dagger \qquad \gamma_M^k = -\gamma_M^{k\dagger} \qquad \gamma_M^5 = \gamma_M^{5\dagger}$$

$$\gamma_M^\nu = -\gamma_M^{\nu*} \qquad \gamma_M^5 = -\gamma_M^{5*}$$

The result is that, at the place of a complex self-conjugated bispinor, which has been constructed out of a left-handed Weyl spinor, one can safely and more suitably employ a real Majorana bispinor: namely,

$$\chi_M(x) = \chi_M^c(x) \qquad \leftrightarrow \qquad \psi_M(x) = S^\dagger \chi_M(x) = \psi_M^*(x)$$

A quite analogous construction can obviously be made, had we started from a right-handed Weyl spinor $\varphi \in D(0, \frac{1}{2})$. Then, the massless Majorana Lagrangian and the ensuing wave field equation take the manifestly real forms

$$\mathcal{L}_M = \tfrac{1}{4}\, \psi_M^\top(x)\, \alpha_M^\nu\, i \overset{\leftrightarrow}{\partial}_\nu \psi_M(x)$$

$$i\overset{\leftrightarrow}{\partial}_M \psi_M(x) = 0 \qquad \psi_M(x) = \psi_M^*(x)$$

$$\alpha_M^\nu = \gamma_M^0\, \gamma_M^\nu \qquad \alpha_M^0 = \mathbb{I} \qquad \beta_M \equiv \gamma_M^0$$

It turns out that, from the manifestly real form of the Majorana Lagrangian, the only relic internal symmetries of the massless Majorana's action integral are the discrete $\mathbb{Z}_2$ symmetry, i.e., $\psi_M(x) \longmapsto -\psi_M(x)$, and a further continuous symmetry under the chiral U(1) group

$$\psi_M(x) \quad \mapsto \quad \psi'_M(x) = \exp\left\{\pm i\theta\, \gamma_M^5\right\} \psi_M(x) \qquad (0 \le \theta < 2\pi) \tag{54}$$

the imaginary unit being convenient to keep the reality of the transformed Majorana bispinor. From Nöther theorem, we get the corresponding real current, which satisfies the continuity equation

$$J_5^\mu(x) = \tfrac{1}{2}\, \psi_M^\top(x)\, \alpha_M^\mu\, i\gamma_M^5\, \psi_M(x) \qquad \partial \cdot J_5(x) = 0 \tag{55}$$

as well as the ensuing conserved pseudo-scalar charge

$$\pm Q_5 = \pm \frac{1}{2} \int d\mathbf{x}\, \psi_M^\top(t, \mathbf{x})\, i\gamma_M^5 \psi_M(t, \mathbf{x}) \qquad \dot{Q}_5 = 0$$

the overall $\pm$ sign being conventional and irrelevant. For anti-commuting Grassmann-valued functions we get

$$\pm Q_5 = \pm \int d\mathbf{x}\, [\, \psi_{M,1}(t, \mathbf{x})\psi_{M,4}(t, \mathbf{x}) - \psi_{M,2}(t, \mathbf{x})\psi_{M,3}(t, \mathbf{x})\,]$$

the integrated quantity being nothing but than $-i\chi^\dagger(x)\chi(x)$ as expected.

The chiral symmetry (54) has been already discussed at the end of Section 2. In Section 8, we will see that the conservation law (55) at the quantum level is violated by an anomaly.

It is interesting to set up the spin-states of the massless Majorana real spinor field, because they sensibly differ from the usual and well-known Dirac spin states (the spin states for Weyl spinor are summarized in Appendix A). Let us start from the constant eigenvectors $\xi_\pm$ of the chiral matrix $\gamma_M^5$ in the Majorana representation which do indeed satisfy by definition

$$\gamma_M^5\, \xi_\pm = \pm\, \xi_\pm$$

Then, for example, we can suitably define the Majorana massless spin-states to be

$$w_r(\mathbf{p}) \equiv p'_M\, \xi_r / 2\wp \qquad (r = \pm,\ p_0 = \wp \equiv |\mathbf{p}|)$$

and the corresponding plane wave functions

$$w_{\mathbf{p},r}(x) \equiv [\,(2\pi)^3 2\wp\,]^{-\frac{1}{2}}\, w_r(\mathbf{p})\, \exp\{-i\wp t + i\mathbf{p} \cdot \mathbf{x}\}$$

The above introduced Majorana massless spin-states are complex and the related plane waves turn out to be a pair of degenerate eigenstates positive frequency or energy solutions of the massless Majorana wave equation $i\partial_\mu \gamma_M^\mu w_{\mathbf{p},r}(x) = 0$ ($r = +, -$). It appears that the pair of orthogonal negative energy spin-states and plane waves functions are nothing but the complex conjugates of the former ones.

The transition to the quantum theory is performed as usual by means of the creation annihilation operators $a_{\mathbf{p},r}$ and $a_{\mathbf{p},s}^{\dagger}$ which satisfy the canonical anti-commutation relations

$$\{a_{\mathbf{p},r}, a_{\mathbf{q},s}\} = 0 = \{a_{\mathbf{p},r}^{\dagger}, a_{\mathbf{q},s}^{\dagger}\} \qquad \{a_{\mathbf{p},r}, a_{\mathbf{q},s}^{\dagger}\} = \delta_{rs}\, \delta(\mathbf{p}-\mathbf{q})$$

with $\mathbf{p}$, $\mathbf{q} \in \mathbb{R}^3$, $r, s = \pm$, so that the operator valued tempered distribution for the Majorana quantum spinor field takes the form

$$\psi_M(x) = \sum_{\mathbf{p},r} \left[ a_{\mathbf{p},r}\, w_{\mathbf{p},r}(x) + a_{\mathbf{p},r}^{\dagger}\, w_{\mathbf{p},r}^*(x) \right] = \psi_M^c(x)$$

where use has been made of the shorthand notation $\sum_{\mathbf{p},r} \equiv \int d\mathbf{p} \sum_{r=\pm}$. Moreover, we readily get the useful expansion for the adjoint quantum fields

$$\overline{\psi}_M(x) = \sum_{\mathbf{q},s} \left[ a_{\mathbf{q},s}^{\dagger}\, \overline{w}_{\mathbf{q},s}(x) + a_{\mathbf{q},s}\, \overline{w}_{\mathbf{q},s}^*(x) \right]$$

in which

$$\overline{w}_{\mathbf{p},r}(x) \equiv [(2\pi)^3 2\wp]^{-\frac{1}{2}} [w_r^{\top}(\mathbf{p})]^* \beta_M \exp\{i\wp t - i\mathbf{p}\cdot\mathbf{x}\}$$

From the orthogonality relations for spin-states and plane waves, we get the expressions for the energy-momentum and helicity observables in terms of normal ordered products of operators

$$P_\mu = \frac{i}{4} \int d\mathbf{x} : \overline{\psi}_M(x)\, \beta_M \overset{\leftrightarrow}{\partial}_\mu \psi_M(x) := \sum_{\mathbf{p},r} p_\mu\, a_{\mathbf{p},r}^{\dagger}\, a_{\mathbf{p},r} \qquad (p_0 = \wp)$$

$$h = \int_{-\infty}^{\infty} dy : \tfrac{1}{2}\, \psi_M(t,y)\, \Sigma_{M,2}\, \psi_M(t,y) := \tfrac{1}{2} \int_{-\infty}^{\infty} dp \left[ a_{p,+}^{\dagger}\, a_{p,+} - a_{p,-}^{\dagger}\, a_{p,-} \right]$$

where a 1D motion along the $Oy$-axis has been referred to, e.g., to select the helicity operator.

$$Q_5 = \tfrac{1}{2} \int d\mathbf{x} : \overline{\psi}_M(t,\mathbf{x})\beta_M \gamma_M^5 \psi(t,\mathbf{x}) := \int d\mathbf{p} \left( a_{\mathbf{p},+}^{\dagger}\, a_{\mathbf{p},+} - a_{\mathbf{p},-}^{\dagger}\, a_{\mathbf{p},-} \right) \equiv \nu_M$$

whence it follows that the above introduced pseudo-scalar charge $Q_5$ is nothing but the quantum counterpart of the Atiah–Singer index for a Majorana spinor field.

Hence, it appears that the 1-particle states $a_{\mathbf{p},\pm}^{\dagger}|0\rangle$ represent neutral Majorana particles with energy-momentum $p^\mu = (\wp, \mathbf{p})$ and positive/negative helicity.

It is worthwhile to remark and gather that, as explicitly shown above, the quanta of the massless spin $\tfrac{1}{2}$ Majorana field are on the light-cone, neutral—i.e., particle and anti-particle actually coincide—and with two polarization states, so that they have really nothing to share with the quanta of a Weyl field, but living on the light-cone, the latter being charged, the particle and antiparticle carrying one single and opposite polarization. In a sense, the mechanical properties of the spin $\tfrac{1}{2}$ massless Majorana quanta are close and similar to those ones of a spin 1 photons, apart from the interactions.

Finally, from the massless Majorana Lagrangian and related wave equation, one can immediately realize that the causal Green function for the Majorana massless neutral spinor field is the very same as for a massless Dirac charged quantum field: namely,

$$\langle 0 | T\, \psi_M(x)\, \overline{\psi}_M(y) | 0 \rangle = S_M(x-y) = \frac{i}{(2\pi)^4} \int d^4 p\, \frac{p\!\!\!/_M}{p^2 + i\varepsilon} \exp\{-ip\cdot x\}$$

the massless limit being smooth, which satisfies $i\partial\!\!\!/_M S_M(x-y) = i\delta(x-y)$.

The second part of this review is devoted to gauge anomalies. Spin states are indispensable for the S matrix elements, but do not play a direct role in the calculation of anomalies. For them we need the fermion determinant and its variations. A formal manipulation of the path integral shows that the

determinant of the Dirac operator for a Dirac spinor is the square of the fermion determinant for a Majorana spinor. One can take the square root of the Dirac determinant as the definition of the fermion determinant (the functional integral) of a Majorana fermion. This formal procedure turns out to be correct, as one can check by comparing with the perturbative approach. In Section 8, we present one of such checks: based on the perturbative results for Weyl fermions and using the representation of a Majorana fermion in terms a Weyl fermion and its Lorentz covariant conjugate (see comment after Equation (12)), we show there that the anomalies for a Majorana fermion are the same as those of a Dirac fermion, with half coefficient.)

## 6. Consistent Gauge Anomalies for Weyl Fermions

As explained in the introduction, anomalies are one of the main topics where Dirac and Weyl fermions split significantly. In the present and forthcoming sections we aim to recalculate all the anomalies (chiral and trace) of Weyl, Dirac and Majorana fermions coupled to gauge potentials, with the basic method of Feynman diagrams. Most results are supposedly well-known. Our purpose is to collect them all in order to highlight their reciprocal relations.

Let us consider the classical action integral for a right-handed Weyl fermion coupled to an external gauge field $V_\mu = V_\mu^a T^a$, $T^a$ being Hermitean generators, $[T^a, T^b] = if^{abc}T^c$ (in the Abelian case $T = 1, f = 0$) in a fundamental representation of e.g., SU(N): namely,

$$S_R[V] = \int d^4x\, i\,\overline{\psi_R}\left(\slashed{\partial} - i\slashed{V}\right)\psi_R. \tag{56}$$

This action is invariant under the gauge transformation $\delta V_\mu = D_\mu \lambda \equiv \partial_\mu \lambda - i[V_\mu, \lambda]$, which implies the conservation of the non-Abelian current $J_{R\mu}^a = \bar{\psi}_R \gamma_\mu T^a \psi_R$, i.e.

$$\nabla \cdot J_R^a \equiv (\partial^\mu \delta^{ac} + f^{abc}V^{b\mu})J_{R\mu}^c = 0. \tag{57}$$

The quantum effective action for this theory is given by the generating functional of the connected Green functions of such currents in the presence of the source $V^{a\mu}$

$$W[V] = W[0] \tag{58}$$
$$+ \sum_{n=1}^{\infty} \frac{i^{n-1}}{n!} \int \prod_{i=1}^{n} d^4x_i\, V^{a_i\mu_i}(x_i)\, \langle 0|\mathcal{T}J_{R\mu_1}^{a_1}(x_1)\dots J_{R\mu_n}^{a_n}(x_n)|0\rangle_c$$

and the full 1-loop 1-point function of $J_{R\mu}^a$ is

$$\langle\!\langle J_{R\mu}^a(x)\rangle\!\rangle = \frac{\delta W[V]}{\delta V^{a\mu}(x)} = \sum_{n=0}^{\infty} \frac{i^n}{n!} \int \prod_{i=1}^{n} d^4x_i\, V^{a_i\mu_i}(x_i)\langle 0|\mathcal{T}J_{R\mu}^a(x)J_{R\mu_1}^{a_1}(x_1)...J_{R\mu_n}^{a_n}(x_n)|0\rangle_c \tag{59}$$

Our purpose here is to calculate the odd parity anomaly of the divergence $\nabla \cdot \langle\!\langle J_R^a \rangle\!\rangle$. As is well-known, the first nontrivial contribution to the anomaly comes from the divergence of the three-point function in the RHS of (59). For simplicity, we will denote it $\langle \partial \cdot J_R\, J_R\, J_R \rangle$. Below we will evaluate it in some detail as a sample for the remaining calculations.

### 6.1. The Calculation

Let us start with dimensional regularisation. The fermion propagator is $\frac{i}{\slashed{p}}$ and the vertex $i\gamma_\mu P_R T^a$. The Fourier transform of the three currents amplitude $\langle J_R\, J_R\, J_R \rangle$ is given by

$$\begin{aligned}
\widetilde{F}_{\mu\lambda\rho}^{(R)abc}(k_1, k_2) &= \int \frac{d^4p}{(2\pi)^4} \text{Tr}\left\{\frac{1}{\slashed{p}}\frac{1-\gamma_5}{2}\gamma_\lambda T^b \frac{1}{\slashed{p}-\slashed{k}_1}\frac{1-\gamma_5}{2}\gamma_\rho T^c \frac{1}{\slashed{p}-\slashed{q}}\frac{1-\gamma_5}{2}\gamma_\mu T^a\right\} \\
&\equiv \text{Tr}(T^a T^b T^c)\widetilde{F}_{\mu\lambda\rho}^{(R)}(k_1, k_2)
\end{aligned} \tag{60}$$

where $q = k_1 + k_2$. The relevant Feynman diagram is shown in Figure 1.

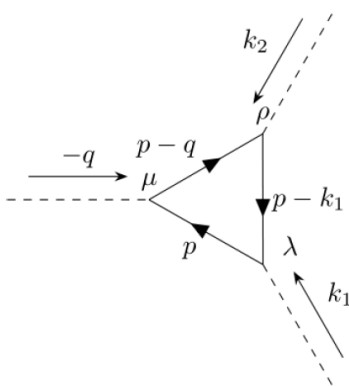

**Figure 1.** The Feynman diagram corresponding to $\widetilde{F}^{(R)abc}_{\mu\lambda\rho}(k_1, k_2)$.

From now on we focus on the Abelian part $\widetilde{F}^{(R)}_{\mu\lambda\rho}(k_1, k_2)$. We dimensionally regularise it by introducing $\delta$ additional dimensions and corresponding momenta $\ell_\mu$, $\mu = 0, \ldots, 3 + \delta$, with the properties

$$\ell\!\!/\, p\!\!/ + p\!\!/\, \ell\!\!/ = 0, \qquad [\ell, \gamma_5] = 0, \qquad p\!\!/^2 = p^2, \qquad \ell\!\!/^2 = -\ell^2$$
$$\mathrm{tr}(\gamma_\mu \gamma_\nu \gamma_\lambda \gamma_\rho \gamma_5) = -2^{2+\frac{\delta}{2}}\, i\, \epsilon_{\mu\nu\lambda\rho},$$

so the relevant expression to be calculated is

$$q^\mu \widetilde{F}^{(R)}_{\mu\lambda\rho}(k_1, k_2) = \int \frac{d^4 p\, d^\delta \ell}{(2\pi)^{4+\delta}} \mathrm{tr}\left\{ \frac{1}{p\!\!/ + \ell\!\!/}\, \frac{1 - \gamma_5}{2}\, \gamma_\lambda\, \frac{1}{p\!\!/ + \ell\!\!/ - k\!\!/_1}\, \frac{1 - \gamma_5}{2}\, \gamma_\rho\, \frac{1}{p\!\!/ + \ell\!\!/ - q\!\!/}\, \frac{1 - \gamma_5}{2}\, q\!\!/ \right\}$$

$$= \int \frac{d^4 p\, d^\delta \ell}{(2\pi)^{4+\delta}} \mathrm{tr}\left\{ \frac{p\!\!/}{p^2 - \ell^2}\, \gamma_\lambda\, \frac{p\!\!/ - k\!\!/_1}{(p - k_1)^2 - \ell^2}\, \gamma_\rho\, \frac{p\!\!/ - q\!\!/}{(p - q)^2 - \ell^2}\, \frac{1 - \gamma_5}{2}\, q\!\!/ \right\}$$

$$\equiv \widetilde{F}^{(R)}_{\lambda\rho}(k_1, k_2, \delta). \tag{61}$$

Now we focus on the odd part and work out the gamma traces:

$$\widetilde{F}^{(R,odd)}_{\lambda\rho}(k_1, k_2, \delta) = -2^{1+\frac{\delta}{2}}\, i\epsilon_{\mu\nu\lambda\rho} \int \frac{d^4 p\, d^\delta \ell}{(2\pi)^{4+\delta}}\, \frac{\left(p^2 q^\mu + (q^2 - 2p\cdot q)p^\mu\right)\left(p^\nu - k_1^\nu\right)}{(p^2 - \ell^2)((p - k_1)^2 - \ell^2)((p - q)^2 - \ell^2)}. \tag{62}$$

Let us write the numerator on the RHS as follows:

$$p^2 q^\mu + (q^2 - 2p\cdot q)p^\mu = -(p^2 - \ell^2)(p - q)^\mu + ((p - q)^2 - \ell^2)p^\mu + \ell^2 q^\mu. \tag{63}$$

Then (62) can be rewritten as

$$\begin{aligned}
\widetilde{F}^{(R,odd)}_{\lambda\rho}(k_1, k_2, \delta) &= -2^{1+\frac{\delta}{2}}\, i\epsilon_{\mu\nu\lambda\rho} \int \frac{d^4 p\, d^\delta \ell}{(2\pi)^{4+\delta}} \left\{ \ell^2 \frac{q^\mu \left(p^\nu - k_1^\nu\right)}{(p^2 - \ell^2)((p - k_1)^2 - \ell^2)((p - q)^2 - \ell^2)} \right. \\
&\qquad \left. + \frac{(q - p)^\mu (p - k_1)^\nu}{((p - k_1)^2 - \ell^2)((p - q)^2 - \ell^2)} + \frac{p^\mu (p - k_1)^\nu}{(p^2 - \ell^2)((p - k_1)^2 - \ell^2)} \right\} \\
&= -2^{1+\frac{\delta}{2}}\, i\epsilon_{\mu\nu\lambda\rho} \int \frac{d^4 p\, d^\delta \ell}{(2\pi)^{4+\delta}} \left\{ \ell^2 \frac{q^\mu \left(p^\nu - k_1^\nu\right)}{(p^2 - \ell^2)((p - k_1)^2 - \ell^2)((p - q)^2 - \ell^2)} \right. \\
&\qquad \left. - \frac{(p - k_2)^\mu p^\nu}{(p^2 - \ell^2)((p - k_2)^2 - \ell^2)} + \frac{p^\mu (p - k_1)^\nu}{(p^2 - \ell^2)((p - k_1)^2 - \ell^2)} \right\}.
\end{aligned} \tag{64, 65}$$

The last two terms do not contribute because of the antisymmetric $\epsilon$ tensor, as one can easily see by introducing a Feynman parameter. The first term can be easily evaluated by introducing two Feynman parameters $x$ and $y$, and making the shift $p \to p + (x+y)k_1 + yk_2$,

$$\widetilde{F}_{\lambda\rho}^{(R,odd)}(k_1,k_2,\delta) = -2^{2+\frac{\delta}{2}} \, i\epsilon_{\mu\nu\lambda\rho} \int_0^1 dx \int_0^{1-x} dy \int \frac{d^4p \, d^\delta\ell}{(2\pi)^{4+\delta}} \ell^2 \frac{q^\mu(p^\nu + (x+y-1)k_1 + yk_2)^\nu)}{(p^2 - \ell^2 + \Delta(x,y))^3} \tag{66}$$

where $\Delta = (x+y)(1-x-y)k_1^2 + y(1-y)k_2^2 + 2y(1-x-y)k_1 \cdot k_2$. Now we make a Wick rotation on the integration momentum, $p^0 \to ip^0$, and the same on $k_1, k_2$ (although we stick to the same symbols).

Then, using

$$\int \frac{d^4p}{(2\pi)^4} \int \frac{d^\delta\ell}{(2\pi)^\delta} \frac{\ell^2}{(p^2 + \ell^2 + \Delta)^3} = -\frac{1}{2(4\pi)^2} \tag{67}$$

and taking the limit $\delta \to 0$, we find

$$\widetilde{F}_{\lambda\rho}^{(R,odd)}(k_1,k_2) = \frac{2}{(4\pi)^2} \epsilon_{\mu\nu\lambda\rho} k_1^\mu k_2^\nu \int_0^1 dx \int_0^{1-x} dy \, (1-x) = \frac{1}{24\pi^2} \epsilon_{\mu\nu\lambda\rho} k_1^\mu k_2^\nu. \tag{68}$$

We must add the cross term (for $\lambda \leftrightarrow \rho$ and $k_1 \leftrightarrow k_2$), so that the total result is

$$\widetilde{F}_{\lambda\rho}^{(R,odd)}(k_1,k_2) + \widetilde{F}_{\rho\lambda}^{(R,odd)}(k_2,k_1) = \frac{1}{12\pi^2} \epsilon_{\mu\nu\lambda\rho} k_1^\mu k_2^\nu. \tag{69}$$

In order to return to configuration space we have to insert this result into (59). We consider here, for simplicity, the Abelian case. We have

$$
\begin{aligned}
\partial^\mu \langle\langle J_{R\mu}(x) \rangle\rangle &= \int \frac{d^4q}{(2\pi)^4} e^{-iqx} (-iq^\mu) \langle\langle \tilde{J}_{R\mu}(q) \rangle\rangle = \frac{i}{2} \int \frac{d^4q}{(2\pi)^4} \int \frac{d^4k_1}{(2\pi)^4} \int \frac{d^4k_2}{(2\pi)^4} \int d^4y \, d^4z \\
&\quad \times q^\mu \, e^{i(k_1 y + k_2 z - qx)} \left( \widetilde{F}_{\lambda\rho}^{(R,odd)}(k_1,k_2) + \widetilde{F}_{\rho\lambda}^{(R,odd)}(k_2,k_1) \right) V^\lambda(y) V^\rho(z).
\end{aligned}
\tag{70}
$$

After a Wick rotation, we can replace (69) inside the integrals

$$
\begin{aligned}
\partial^\mu \langle\langle J_{R\mu}(x) \rangle\rangle &= -\frac{1}{24\pi^2} \int \frac{d^4q \, d^4k_1 \, d^4k_2}{(2\pi)^{12}} \int d^4y \, d^4z \, e^{i(qx - k_1 y - k_2 z)} \delta(q - k_1 - k_2) \epsilon_{\mu\nu\lambda\rho} k_1^\mu k_2^\nu \, V^\lambda(y) V^\rho(z) \\
&= -\frac{1}{24\pi^2} \int \frac{d^4k_1}{(2\pi)^4} \int \frac{d^4k_2}{(2\pi)^4} \int d^4y \, d^4z \, e^{ik_1(x-y)} e^{-ik_2(x-z)} \epsilon_{\mu\nu\lambda\rho} \partial^\mu V^\lambda(y) \partial^\nu V^\rho(z) \\
&= \frac{1}{24\pi^2} \epsilon_{\mu\nu\lambda\rho} \partial^\mu V^\nu(x) \partial^\lambda V^\rho(x).
\end{aligned}
\tag{71}
$$

The same result can be obtained with the Pauli–Villars regularisation, see Appendix B.

### 6.2. Comments

Equation (71) is the consistent gauge anomaly of a right-handed Weyl fermion coupled to an Abelian vector field $V_\mu(x)$. It is well known that the consistent anomaly (71) destroys the consistency of the Abelian gauge theory. As a matter of fact the Lorentz invariant quantum theory of a gauge vector field unavoidably involves a Fock space of states with indefinite norm. Now, in order to select a physical Hilbert subspace of the Fock space, a subsidiary condition is necessary. In the Abelian case, when the fermion current satisfies the continuity equation the equations of motion lead to $\Box(\partial \cdot V) = 0$, so that a subspace of states of non-negative norm can be selected through the auxiliary condition

$$\partial \cdot V^{(-)}(x) | \text{phys} \rangle = 0$$

$V^{(-)}(x)$ being the annihilation operator, the positive frequency part of a d'Alembert quantum field. On the contrary , in the present chiral model we find

$$\Box(\partial \cdot V) = -\frac{1}{3}\left(\frac{1}{4\pi}\right)^2 F_*^{\mu\nu} F_{\mu\nu} \neq 0$$

in such a manner that nobody knows how to select a physical subspace of states with non-negative norm, if any, where a unitary restriction of the collision operator $S$ could be defined.

Another way of seeing the problem created by the consistent anomaly is to remark that, for instance, $J_{R\mu}$ couples minimally to $V^\mu$ at the fermion-fermion-gluon vertex. Unitarity and renormalisability rely on the Ward identity that guarantees current conservation at any such vertex. This is impossible in the presence of a consistent anomaly.

The consistent anomaly in the non-Abelian case would require the calculation of at least the four current correlators, but it can be obtained in a simpler way from the Abelian case using the Wess–Zumino consistency conditions. In the non-Abelian case the three-point correlators are multiplied by

$$\mathrm{Tr}(T^a T^b T^c) = \frac{1}{2}\mathrm{Tr}(T^a[T^b, T^c]) + \frac{1}{2}\mathrm{Tr}(T^a\{T^b, T^c\}) = f^{abc} + d^{abc} \tag{72}$$

where the normalisation used is $\mathrm{Tr}(T^a T^b) = 2\delta^{ab}$. Since the three-point function is the sum of two equal pieces with $\lambda \leftrightarrow \rho, k_1 \leftrightarrow k_2$, the first term in the RHS of (72) drops out and only the second remains. For the right-handed current $J_{R\mu}^a$ we have

$$\nabla \cdot \langle\!\langle J_R^a \rangle\!\rangle = \frac{1}{24\pi^2}\varepsilon_{\mu\nu\lambda\rho}\mathrm{Tr}\left[T^a\partial^\mu\left(V^\nu\partial^\lambda V^\rho + \frac{i}{2}V^\nu V^\lambda V^\rho\right)\right]. \tag{73}$$

The previous results are well-known (for a short introduction to consistency and covariance applied to anomalies, see Appendix C). However, they do not tell the whole story about gauge anomalies in a theory of Weyl fermions. To delve into this we have to enlarge the parameter space by coupling the fermions to an additional potential, namely to an axial vector field.

## 7. The V-A Anomalies

The action of a Dirac fermion coupled to a vector $V_\mu$ and an axial potential $A_\mu$ (for simplicity we consider only the Abelian case) is

$$S[V, A] = \int d^4x\, i\,\overline{\psi}\left(\slashed{\partial} - i\slashed{V} - i\slashed{A}\gamma_5\right)\psi. \tag{74}$$

The generating functional of the connected Green functions is

$$\begin{aligned}
W[V, A] &= W[0, 0] + \sum_{n,m=1}^{\infty}\frac{i^{n+m-1}}{n!m!}\int\prod_{i=1}^{n}d^4x_i\, V^{\mu_i}(x_i)\prod_{j=1}^{m}d^4y_j\, A^{\nu_j}(y_j) \\
&\quad \times \langle 0|\mathcal{T}J_{\mu_1}(x_1)\ldots J_{\mu_n}(x_n)J_{5\nu_1}(y_1)\ldots J_{5\nu_m}(x_m)|0\rangle_c.
\end{aligned} \tag{75}$$

We can extract the full one-loop one-point function for two currents: the vector current $J_\mu = \overline{\psi}\gamma_\mu\psi$

$$\begin{aligned}
\langle\!\langle J_\mu(x)\rangle\!\rangle &= \frac{\delta W[V, A]}{\delta V^\mu(x)} = \sum_{n,m=0}^{\infty}\frac{i^{n+m}}{n!m!}\int\prod_{i=1}^{n}d^4x_i\, V^{\mu_i}(x_i)\prod_{j=1}^{m}d^4y_j\, A^{\nu_j}(y_j) \\
&\quad \times \langle 0|\mathcal{T}J_\mu(x)J_{\mu_1}(x_1)\ldots J_{\mu_n}(x_n)J_{5\nu_1}(y_1)\ldots J_{5\nu_m}(x_m)|0\rangle_c
\end{aligned} \tag{76}$$

and the axial current $J_\mu = \bar{\psi}\gamma_\mu\gamma_5\psi$

$$
\begin{aligned}
\langle\langle J_{5\mu}(x)\rangle\rangle &= \frac{\delta W[V,A]}{\delta A^\mu(x)} = \sum_{n,m=0}^\infty \frac{i^{n+m}}{n!m!} \int \prod_{i=1}^n d^4x_i \, V_{\mu_i}(x_i) \prod_{j=1}^m d^4y_j A^{\nu_j}(y_j) \\
&\quad \times \langle 0|\mathcal{T}J_{5\mu}(x)J_{\mu_1}(x_1)\dots J_{\mu_n}(x_n)J_{5\nu_1}(y_1)\dots J_{5\nu_m}(x_m)|0\rangle_c.
\end{aligned}
\tag{77}
$$

These currents are conserved except for possible anomaly contributions. The aim of this section is to study the continuity equations for these currents, that is to compute the 4-divergences of the correlators on the RHS of (76) and (77). For the same reason explained above we focus on the three current correlators: they are all we need in the Abelian case (and the starting point to compute the full anomaly expression by means of the Wess–Zumino consistency conditions in the non-Abelian case). For $\partial \cdot J(x)$, the first relevant contributions are

$$
\begin{aligned}
\partial^\mu\langle\langle J_\mu(x)\rangle\rangle &= -\Big(\frac{1}{2}\int d^4x_1 d^4x_2 V^{\mu_1}(x_1)V^{\mu_2}(x_2)\partial^\mu\langle 0|\mathcal{T}J_\mu(x)J_{\mu_1}(x_1)J_{\mu_2}(x_2)|0\rangle \\
&\quad + \int d^4x_1 d^4y_1 V^{\mu_1}(x_1)A^{\nu_1}(y_1)\partial^\mu\langle 0|\mathcal{T}J_\mu(x)J_{\mu_1}(x_1)J_{5\nu_1}(y_1)|0\rangle \\
&\quad + \frac{1}{2}\int d^4y_1 d^4y_2 A^{\nu_1}(y_1)A^{\nu_2}(y_2)\partial^\mu\langle 0|\mathcal{T}J_\mu(x)J_{5\nu_1}(y_1)J_{5\nu_2}(y_2)|0\rangle\Big)
\end{aligned}
\tag{78}
$$

and for $\partial^\mu J_{5\mu}(x)$

$$
\begin{aligned}
\partial^\mu\langle\langle J_{5\mu}(x)\rangle\rangle &= -\Big(\frac{1}{2}\int d^4x_1 d^4x_2 V^{\mu_1}(x_1)V^{\mu_2}(x_2)\partial^\mu\langle 0|\mathcal{T}J_{5\mu}(x)J_{\mu_1}(x_1)J_{\mu_2}(x_2)|0\rangle \\
&\quad + \int d^4x_1 d^4y_1 V^{\mu_1}(x_1)A^{\nu_1}(y_1)\partial^\mu\langle 0|\mathcal{T}J_{5\mu}(x)J_{\mu_1}(x_1)J_{5\nu_1}(y_1)|0\rangle \\
&\quad + \frac{1}{2}\int d^4y_1 d^4y_2 A^{\nu_1}(y_1)A^{\nu_2}(y_2)\partial^\mu\langle 0|\mathcal{T}J_{5\mu}(x)J_{5\nu_1}(y_1)J_{5\nu_2}(y_2)|0\rangle\Big).
\end{aligned}
\tag{79}
$$

Since we are interested in odd parity anomalies, the only possible contribution to (78) is from the term in the second line, which we denote concisely $\langle\partial\cdot J\, J\, J_5\rangle$. As for (79), there are two possible contributions from the first and third lines, i.e., $\langle\partial\cdot J_5\, J\, J\rangle$ and $\langle\partial\cdot J_5\, J_5\, J_5\rangle$. Below we report the results for the corresponding amplitudes, obtained with dimensional regularisation.

The amplitude for $\langle\partial\cdot J_5\, J\, J\rangle$ is

$$
q^\mu\widetilde{F}^{(5)}_{\mu\lambda\rho}(k_1,k_2) = \int \frac{d^4p\,d^\delta\ell}{(2\pi)^{4+\delta}}\mathrm{tr}\left\{\frac{1}{\not p + \not\ell}\gamma_\lambda\frac{1}{\not p + \not\ell - \not k_1}\gamma_\rho\frac{1}{\not p + \not\ell - \not q}\not q\gamma_5\right\}.
\tag{80}
$$

The relevant Feynman diagram is shown in Figure 2.

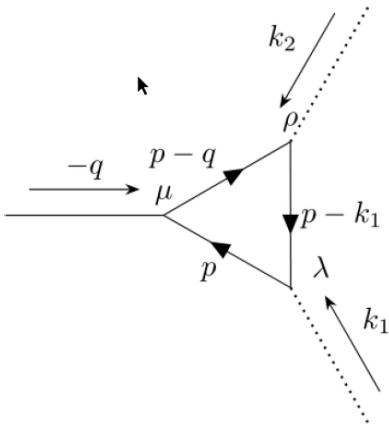

**Figure 2.** The Feynman diagram corresponding to $\widetilde{F}^{(5)}_{\mu\lambda\rho}(k_1,k_2)$.

Adding the cross contribution one gets

$$q^\mu \left( \widetilde{F}^{(5)}_{\mu\lambda\rho}(k_1, k_2) + T^{(5)}_{\mu\rho\lambda}(k_2, k_1) \right) = \frac{1}{2\pi^2} \epsilon_{\mu\nu\lambda\rho} k_1^\mu k_2^\nu. \tag{81}$$

The amplitude for $\langle \partial \cdot J_5 \, J_5 \, J_5 \rangle$ is given by

$$
\begin{aligned}
q^\mu \widetilde{F}^{(555)}_{\mu\lambda\rho}(k_1, k_2) &= \int \frac{d^4 p \, d^\delta \ell}{(2\pi)^{4+\delta}} \mathrm{tr} \left\{ \frac{1}{\slashed{p} + \slashed{\ell}} \gamma_\lambda \gamma_5 \frac{1}{\slashed{p} + \slashed{\ell} - \slashed{k}_1} \gamma_\rho \gamma_5 \frac{1}{\slashed{p} + \slashed{\ell} - \slashed{q}} \slashed{q} \gamma_5 \right\} \\
&= -2^{2+\frac{\delta}{2}} i\epsilon_{\mu\nu\lambda\rho} \int \frac{d^4 p \, d^\delta \ell}{(2\pi)^{4+\delta}} \, \ell^2 \, \frac{q^\mu(3p^\nu - k_1^\nu)}{(p^2 - \ell^2)((p - k_1)^2 - \ell^2)((p - q)^2 - \ell^2)} \\
&\quad - \int \frac{d^4 p \, d^\delta \ell}{(2\pi)^{4+\delta}} \frac{\mathrm{tr}(\slashed{q} \, \slashed{p} \, \gamma_\lambda (\slashed{p} - \slashed{k}_1) \gamma_\rho (\slashed{p} - \slashed{q}) \gamma_5)}{(p^2 - \ell^2)((p - k_1)^2 - \ell^2)((p - q)^2 - \ell^2)}
\end{aligned} \tag{82}
$$

The first line in the last expression, after introducing the Feynman parameters $x$ and $y$ and shifting $p$ as usual, yields a factor $\int_0^1 dx \int_0^{1-x} dy \, (1 - 3x) = 0$, so it vanishes. The last line is $2 \times \widetilde{F}^{(R,odd)}_{\lambda\rho}(k_1, k_2, \delta)$, cf. (61) and (62). Therefore, using (69), we get

$$q^\mu \left( \widetilde{F}^{(555)}_{\mu\lambda\rho}(k_1, k_2) + \widetilde{F}^{(555)}_{\mu\rho\lambda}(k_2, k_1) \right) = \frac{1}{6\pi^2} \epsilon_{\mu\nu\lambda\rho} k_1^\mu k_2^\nu. \tag{83}$$

Finally the amplitude for $\langle \partial \cdot J \, J \, J_5 \rangle$ is

$$q^\mu \widetilde{F}^{(5')}_{\mu\lambda\rho}(k_1, k_2) = \int \frac{d^4 p \, d^\delta \ell}{(2\pi)^{4+\delta}} \mathrm{tr} \left\{ \frac{1}{\slashed{p} + \slashed{\ell}} \gamma_\lambda \frac{1}{\slashed{p} + \slashed{\ell} - \slashed{k}_1} \gamma_\rho \gamma_5 \frac{1}{\slashed{p} + \slashed{\ell} - \slashed{q}} \slashed{q} \right\} = 0. \tag{84}$$

All the above results have been obtained also with PV regularisation.

Plugging in these results in (76) and (77) we find

$$\partial^\mu \langle\!\langle J_\mu(x) \rangle\!\rangle = 0 \tag{85}$$

and

$$\partial^\mu \langle\!\langle J_{5\mu}(x) \rangle\!\rangle = \frac{1}{4\pi^2} \epsilon_{\mu\nu\lambda\rho} \left( \partial^\mu V^\nu(x) \partial^\lambda V^\rho(x) + \frac{1}{3} \partial^\mu A^\nu(x) \partial^\lambda A^\rho(x) \right) \tag{86}$$

which is Bardeen's result [19], in the Abelian case. From (86) we can derive the covariant chiral anomaly by setting $A_\mu = 0$, then

$$\partial^\mu \langle\!\langle J_{5\mu}(x) \rangle\!\rangle = \frac{1}{4\pi^2} \epsilon_{\mu\nu\lambda\rho} \partial^\mu V^\nu(x) \partial^\lambda V^\rho(x). \tag{87}$$

Of course this is nothing but (81). For the $J_{5\mu}(x)$ current is obtained by differentiating the action with respect to $A_\mu(x)$ and its divergence leads to the covariant anomaly.

*Some Conclusions*

Let us recall that in the collapsing limit $V \to V/2$, $A \to V/2$ in the action (74) we recover the theory of a right-handed Weyl fermion (with the addition of a free left-handed part, as explained at length above). Now $J_\mu(x) = J_{R\mu}(x) + J_{L\mu}(x)$ and $J_{5\mu}(x) = J_{R\mu}(x) - J_{L\mu}(x)$. In the collapsing limit we find

$$\partial^\mu \langle\!\langle J^{(cs)}_{R\mu}(x) \rangle\!\rangle = \frac{1}{24\pi^2} \epsilon_{\mu\nu\lambda\rho} \partial^\mu V^\nu(x) \partial^\lambda V^\rho(x). \tag{88}$$

Similarly

$$\partial^\mu \langle\langle J^{(cs)}_{L\mu}(x)\rangle\rangle = -\frac{1}{24\pi^2}\epsilon_{\mu\nu\lambda\rho}\,\partial^\mu V^\nu(x)\partial^\lambda V^\rho(x). \tag{89}$$

These are the consistent right and left gauge anomalies—the label $^{(cs)}$ stands for consistent, to be distinguished from the covariant anomaly. As a matter of fact, application of the same chiral current splitting to the covariant anomaly of Equation (87) yields instead

$$\partial^\mu \langle\langle J^{(cv)}_{R\mu}(x)\rangle\rangle = \frac{1}{8\pi^2}\epsilon_{\mu\nu\lambda\rho}\,\partial^\mu V^\nu(x)\partial^\lambda V^\rho(x) \tag{90}$$

and

$$\partial^\mu \langle\langle J^{(cv)}_{L\mu}(x)\rangle\rangle = -\frac{1}{8\pi^2}\epsilon_{\mu\nu\lambda\rho}\,\partial^\mu V^\nu(x)\partial^\lambda V^\rho(x). \tag{91}$$

The label $^{(cv)}$ stands for covariant, and it is in order to tell apart these anomalies from the previous consistent ones. The two cases should not be confused: the consistent anomalies appears in the divergence of a current minimally coupled in the action to the vector potential $V_\mu$. They represent the response of the effective action under a gauge transform of $V_\mu$, which is supposed to propagate in the internal lines of the corresponding gauge theory. The covariant anomalies represent the response of the effective action under a gauge transform of the external axial current $A_\mu$.

It goes without saying that, both for right and left currents in the collapsing limit, in the non-Abelian case the consistent anomaly takes the form (73), while the covariant one reads

$$\nabla \cdot \langle\langle J^a_R(x)\rangle\rangle = \frac{1}{32\pi^2}\epsilon_{\mu\nu\lambda\rho}\operatorname{Tr}\left(T^a F^{\mu\nu}(x)F^{\lambda\rho}(x)\right) \tag{92}$$

where $F_{\mu\nu}(x) = F^a_{\mu\nu}(x)T^a$ denotes the usual non-Abelian field strength. At first sight the above distinction between covariant and consistent anomalies for Weyl fermion may appear to be academic. After all, if a theory has a consistent anomaly it is ill-defined and the existence of a covariant anomaly may sound irrelevant. However this distinction becomes interesting in some non-Abelian cases since the non-Abelian consistent anomaly is proportional to the tensor $d^{abc}$. Now for most simple gauge groups (except $SU(N)$ for $N \geq 3$) this tensor vanishes identically. In such cases the consistent anomaly is absent and so the covariant anomaly becomes significant.

## 8. The Case of Majorana Fermions

As we have seen above, Majorana fermions are defined by the condition

$$\Psi = \widehat{\Psi}, \qquad \text{where} \qquad \widehat{\Psi} = \gamma_0 C \Psi^*. \tag{93}$$

Let $\psi_R = P_R\psi$ be a generic Weyl fermion. We have

$$P_R\psi_R = \psi_R \qquad P_L\widehat{\psi_R} = \widehat{\psi_R}$$

i.e., $\widehat{\psi_R}$ is left-handed. We have already remarked that $\psi_M = \psi_R + \widehat{\psi_R}$ is a Majorana fermion and any Majorana fermion can be represented in this way. Using this correspondence one can transfer the

results for Weyl fermions to Majorana fermions[9]. The vector current is defined by $J_M^\mu = \bar{\psi}_M \gamma^\mu \psi_M$ and the axial current by $J_{5M}^\mu = \bar{\psi}_M \gamma^\mu \gamma_5 \psi_M$. We can write

$$J_M^\mu(x) = \overline{\psi_R}(x)\gamma^\mu \psi_R(x) + \overline{\widehat{\psi_R}}(x)\gamma^\mu \widehat{\psi_R}(x) \equiv J_R^\mu(x) + J_L^\mu(x) \tag{94}$$

and

$$J_{5M}^\mu(x) = \overline{\psi_R}(x)\gamma^\mu \psi_R(x) - \overline{\widehat{\psi_R}}(x)\gamma^\mu \widehat{\psi_R}(x) \equiv J_R^\mu(x) - J_L^\mu(x). \tag{95}$$

Using (88) and (89) one concludes that, as far as the consistent anomaly is concerned,

$$\partial_\mu \langle\langle J_M^\mu(x) \rangle\rangle = 0 \tag{96}$$

This shows the consistency of our procedure, for one can show that, in general, $J_L(x) = -J_R(x)$, and $J_M(x) = 0$, as it should be for a Majorana fermion. On the other hand for the axial current we have

$$\partial_\mu \langle\langle J_{5M}^\mu(x) \rangle\rangle = \frac{1}{8\pi^2} \epsilon_{\mu\nu\lambda\rho} \partial^\mu V^\nu(x) \partial^\lambda V^\rho(x) \tag{97}$$

where the naïve sum has been divided by 2, because the two contributions come from the same degrees of freedom (which are half those of a Dirac fermion). From these results we see that, apart from the coefficient difference, the anomalies of a massless Majorana fermion are the same as those of a massless Dirac fermion. These obviously descend from the fact that both Dirac and Majorana fermions contain two opposite chiralities, at variance with a Weyl fermion, which is characterized by one single chirality.

## 9. Relation between Chiral and Trace Gauge Anomalies

There exists a strict relation between chiral gauge anomalies and trace anomalies in a theory of fermions coupled to a vector (and axial) gauge potential. This section is devoted to analysing this relation. When the background fields are not only $V_\mu$ and $A_\mu$, but also a non-trivial tensor-axial metric $G_{\mu\nu} = g_{\mu\nu} + \gamma_5 f_{\mu\nu}$, see [17], the generating function must include two energy-momentum tensors, which in the flat-space limit take the Belifante–Rosenfeld symmetric form

$$T^{\mu\nu} = -\frac{i}{4}\left(\bar{\psi}\gamma^\mu \overset{\leftrightarrow}{\partial}^\nu \psi + \mu \leftrightarrow \nu\right), \tag{98}$$

and

$$T_5^{\mu\nu} = \frac{i}{4}\left(\bar{\psi}\gamma_5\gamma^\mu \overset{\leftrightarrow}{\partial}^\nu \psi + \mu \leftrightarrow \nu\right). \tag{99}$$

The quantities we are interested in here are, in particular, the 1-loop VEVs $\langle\langle T_{\mu\nu}(x) \rangle\rangle$ and $\langle\langle T_{5\mu\nu}(x) \rangle\rangle$ when $h_{\mu\nu} = f_{\mu\nu} = 0$ : namely,

$$\begin{aligned}
\langle\langle T_{\mu\nu}(x) \rangle\rangle &= \sum_{r,s=0}^{\infty} \frac{i^{r+s}}{2\,r!s!} \int \prod_{l=1}^{r} d^4 x_l V^{\sigma_l}(x_l) \prod_{k=1}^{s} d^4 y_k A^{\tau_k}(y_k) \tag{100} \\
&\times\ \langle 0| \mathcal{T}\, T_{\mu\nu}(x) J_{\sigma_1}(x_1)\dots J_{\sigma_r}(x_r) J_{5\tau_1}(y_1)\dots J_{5\tau_s}(y_s)|0\rangle
\end{aligned}$$

---

[9]  The converse is not true. It is impossible to reconstruct Weyl fermion anomalies from those of a massless Majorana fermion.

and

$$
\langle\langle T_5^{\mu\nu}(x)\rangle\rangle \;=\; \sum_{r,s=0}^{\infty} \frac{i^{r+s}}{2\,r!s!} \int \prod_{l=1}^{r} d^4 x_l\, V^{\sigma_l}(x_l) \prod_{k=1}^{s} d^4 y_k\, A^{\tau_k}(y_k) \tag{101}
$$
$$
\times \quad \langle 0|\mathcal{T}\, T_5^{\mu\nu}(x) J_{\sigma_1}(x_1)\dots J_{\sigma_r}(x_r) J_{5\tau_1}(y_1)\dots J_{5\tau_s}(y_s)|0\rangle
$$

of which we will compute the trace, i.e., contraction, over the indices $\mu$ and $\nu$. Since we are interested in odd parity anomalies, the first nontrivial contributions come from the three-point correlators (i.e., $r+s=2$). Denoting by $t, t_5$ the traces of $T_{\mu\nu}, T_{5\mu\nu}$, the relevant correlators are $\langle t\, J\, J_5\rangle$ for (100), and $\langle t_5\, J\, J\rangle$, $\langle t_5\, J_5\, J_5\rangle$ for (101). We claim that they are simply related to $\langle \partial\cdot J\, J\, J_5\rangle$, $\langle \partial\cdot J_5\, J\, J\rangle$ and $\langle \partial\cdot J_5\, J_5\, J_5\rangle$, respectively.

We will also need

$$
T_R^{\mu\nu}(x) = \tfrac{1}{2}\left[\, T^{\mu\nu}(x) + T_5^{\mu\nu}(x)\,\right]
$$
$$
\tag{102}
$$
$$
T_L^{\mu\nu}(x) = \tfrac{1}{2}\left[\, T^{\mu\nu}(x) - T_5^{\mu\nu}(x)\,\right]
$$

together with the one-loop 1-point function

$$
\langle\langle T_{R,L}^{\mu\nu}(x)\rangle\rangle = \sum_{r=0}^{\infty} \frac{i^r}{r!} \int \prod_{l=1}^{r} d^4 x_l\, V^{\sigma_l}(x_l) \langle 0|\mathcal{T}\, T_{R,L}^{\mu\nu}(x)\, J_{R,L\sigma_l}(x_l)|0\rangle. \tag{103}
$$

## 9.1. Difference between Gauge and Trace Anomaly

Let us start from the case of the right-handed fermion. The correlator is, symbolically, $\langle t_R\, J_R\, J_R\rangle$, i.e., $\langle 0|\mathcal{T}\, T_{R\mu}^{\mu}(x)\, J_{R\lambda(y)}\, J_{R\rho(z)}|0\rangle$, its Fourier transform being given by

$$
\widetilde{F}_{\mu\lambda\rho}^{(R)\mu}(k_1, k_2) \;=\; \frac{1}{4}\int \frac{d^4 p}{(2\pi)^4}\, \mathrm{tr}\left[\frac{1}{\not p}\frac{1-\gamma_5}{2}\gamma_\lambda \frac{1}{\not p - \not k_1}\frac{1-\gamma_5}{2}\gamma_\rho \frac{1}{\not p - \not k_1 - \not k_2}\frac{1-\gamma_5}{2}(2\not p - \not q)\right]. \tag{104}
$$

The relevant Feynman diagram is shown in Figure 3.

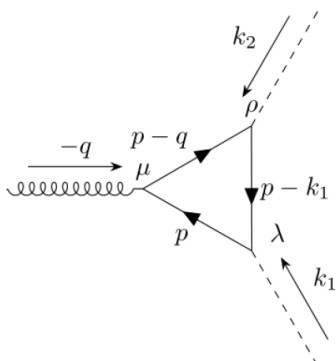

**Figure 3.** The Feynman diagram corresponding to $\widetilde{F}_{\mu\lambda\rho}^{(R)\mu}(k_1, k_2)$.

The difference with respect to the Fourier transform of $\langle \partial \cdot J_R J_R J_R \rangle$—see Equation (61)—apart from the factor $\frac{1}{4}$, is the $(2p - q)$ factor in the RHS, instead of $q$. The relevant difference is therefore twice

$$
\begin{aligned}
\Delta \widetilde{F}^{(R)\mu}_{\mu\lambda\rho}(k_1, k_2) &= \frac{1}{4} \int \frac{d^4 p}{(2\pi)^4} \text{tr} \left[ \frac{1}{\not{p}} \frac{1 - \gamma_5}{2} \gamma_\lambda \frac{1}{\not{p} - \not{k}_1} \frac{1 - \gamma_5}{2} \gamma_\rho \frac{1}{\not{p} - \not{k}_1 - \not{k}_2} \frac{1 - \gamma_5}{2} \not{p} \right] \\
&= \frac{1}{4} \int \frac{d^4 p \, d^\delta \ell}{(2\pi)^{4+\delta}} \text{tr} \left[ \frac{1}{\not{p} + \not{\ell}} \frac{1 - \gamma_5}{2} \gamma_\lambda \frac{1}{\not{p} + \not{\ell} - \not{k}_1} \frac{1 - \gamma_5}{2} \gamma_\rho \frac{1}{\not{p} + \not{\ell} - \not{q}} \frac{1 - \gamma_5}{2} (\not{p} + \not{\ell}) \right] \\
&= \frac{1}{4} \int \frac{d^4 p \, d^\delta \ell}{(2\pi)^{4+\delta}} \frac{\text{tr} \left[ \gamma_\lambda (\not{p} - \not{k}_1) \gamma_\rho (\not{p} - \not{q}) \frac{1 - \gamma_5}{2} \right]}{((p - k_1)^2 - \ell^2)((p - q)^2 - \ell^2)}.
\end{aligned}
$$
(105)

(106)

We can now replace $p \to p + k_1$

$$
\Delta \widetilde{F}^{(R)\mu}_{\mu\lambda\rho}(k_1, k_2) = \frac{i}{4} \int \frac{d^4 p \, d^\delta \ell}{(2\pi)^{4+\delta}} \frac{\text{tr} \left[ \gamma_\lambda \not{p} \gamma_\rho (\not{p} - \not{k}_2) \frac{1 - \gamma_5}{2} \right]}{(p^2 - \ell^2)((p - k_2)^2 - \ell^2)}.
$$
(107)

The odd part vanishes by symmetry.

If we consider instead the amplitude for $\langle \partial \cdot J_5 J J \rangle$, (80), the result does not change. In that case, for the odd part we get

$$
\begin{aligned}
\Delta \widetilde{F}^{(5)\mu}_{\mu\lambda\rho}(k_1, k_2) &\sim \int \frac{d^4 p \, d^\delta \ell}{(2\pi)^{4+\delta}} \text{tr} \left( \gamma_\lambda \frac{\not{p} + \not{\ell} - \not{k}_1}{(p - k_1)^2 - \ell^2} \gamma_\rho \frac{\not{p} + \not{\ell} - \not{q}}{(p - q)^2 - \ell^2} \gamma_5 \right) \\
&= \int \frac{d^4 p \, d^\delta \ell}{(2\pi)^{4+\delta}} \frac{(-\ell^2 \text{tr}(\gamma_\lambda \gamma_\rho \gamma_5) + \text{tr} \left( \gamma_\lambda (\not{p} - \not{k}_1) \gamma_\rho (\not{p} - \not{q}) \gamma_5 \right))}{((p - k_1)^2 - \ell^2)((p - q)^2 - \ell^2)}.
\end{aligned}
$$
(108)

The first term in the numerator vanishes. The rest can be rewritten as

$$
\Delta \widetilde{F}^{(5)\mu}_{\mu\lambda\rho}(k_1, k_2) \sim \int \frac{d^4 p \, d^\delta \ell}{(2\pi)^{4+\delta}} \frac{\text{tr} \left( \gamma_\lambda \not{p} \gamma_\rho (\not{p} - \not{k}_2) \gamma_5 \right)}{(p^2 - \ell^2)((p - k_2)^2 - \ell^2)} = 0
$$
(109)

for the same reason as above. In the same way one can easily prove that

$$
\Delta \widetilde{F}^{(5')\mu}_{\mu\lambda\rho}(k_1, k_2) = \Delta \widetilde{F}^{(5'')\mu}_{\mu\lambda\rho}(k_1, k_2) = 0.
$$
(110)

In conclusion, the amplitude for the chiral anomalies and those for the trace anomalies due to couplings with gauge fields are rigidly related, the corresponding coefficients exhibiting a fixed ratio, i.e., the former are minus four times the latter.

### 9.2. Trace Anomalies Due to a Gauge Field

Using the above results, which say that the difference between $(-4) \times$[the divergence] and the trace anomaly is null, we can immediately deduce the corresponding consistent gauge trace anomalies, using (100), (101) and (103), viz.,

$$
\langle\langle T^{(cs)\mu}_{R\mu}(x) \rangle\rangle = -\frac{1}{48\pi^2} \epsilon_{\mu\nu\lambda\rho} \partial^\mu V^\nu(x) \partial^\lambda V^\rho(x).
$$
(111)

As for $T_{L\mu}(x)$, it carries the consistent anomaly

$$
\langle\langle T^{(cs)\mu}_{L\mu}(x) \rangle\rangle = \frac{1}{48\pi^2} \epsilon_{\mu\nu\lambda\rho} \partial^\mu V^\nu(x) \partial^\lambda V^\rho(x).
$$
(112)

On the other hand, in the $V - A$ framework we find

$$
\langle\langle T^\mu_\mu(x) \rangle\rangle = 0
$$
(113)

and

$$\partial_\mu \langle\langle J_5^\mu(x) \rangle\rangle = -\frac{1}{16\pi^2} \epsilon_{\mu\nu\lambda\rho} \left( \partial^\mu V^\nu(x) \partial^\lambda V^\rho(x) + \frac{1}{3} \partial^\mu A^\nu(x) \partial^\lambda A^\rho(x) \right). \tag{114}$$

From (114) we can derive the covariant chiral anomaly for a Dirac fermion by setting $A_\mu = 0$, then

$$\langle\langle T_{5\mu}^{(cv)\mu}(x) \rangle\rangle = \frac{1}{16\pi^2} \epsilon_{\mu\nu\lambda\rho}\, \partial^\mu V^\nu(x) \partial^\lambda V^\rho(x). \tag{115}$$

From this we can derive the covariant (invariant) trace anomaly for a right-handed

$$\langle\langle T_{R\mu}^{(cv)\mu}(x) \rangle\rangle = -\frac{1}{16\pi^2} \epsilon_{\mu\nu\lambda\rho}\, \partial^\mu V^\nu(x) \partial^\lambda V^\rho(x) \tag{116}$$

and left-handed Weyl fermion

$$\langle\langle T_{L\mu}^{(cv)\mu}(x) \rangle\rangle = \frac{1}{16\pi^2} \epsilon_{\mu\nu\lambda\rho}\, \partial^\mu V^\nu(x) \partial^\lambda V^\rho(x). \tag{117}$$

*9.3. Gauge Anomalies and Diffeomorphisms*

In this review we have not considered diffeomorphisms. Nevertheless a devil's accountant could argue that there might be violation of diffeomorphism invariance in a fermionic system coupled to gauge fields, due to the presence of the gauge fields themselves. In order to see this one has to consider three point correlators involving the divergence of the energy momentum tensor and two currents. More precisely, odd parity anomalies could appear in the following amplitudes: $\langle \partial \cdot T_R\, J_R\, J_R \rangle$, in the right-handed fermion case, or $\langle \partial \cdot T_5\, J\, J \rangle$, $\langle \partial \cdot T\, J_5\, J \rangle$, $\langle \partial \cdot T_5\, J_5\, J_5 \rangle$ in the $V - A$ case. They can be computed with the same methods as above, and here, for brevity, we limit ourselves to record the final results: they all vanish.

## 10. Conclusions

The purpose of this review was to highlight some subtle aspects of the physics of Weyl fermions, as opposed in particular to massless Majorana spinors. To this end we have decided not to resort to powerful non-perturbative methods, like the Seeley–Schwinger–DeWitt method, which would require a demanding introduction. Rather, we have used the simple Feynman diagram technique. In doing so, we have focused on two aims. The first one is to justify the method of computing the effective action for a Weyl fermion coupled to gauge potentials, which requires the presence of free fermions of opposite chirality, in such a way as to produce the effective kinetic operator of Equation (30). We have shown that, notwithstanding the presence of fermions of both chiralities, no mass term can arise as a consequence of quantum corrections. As a by-product we were led to the conclusion that, while the Pauli–Villars regularisation is a perfectly available and useful tool for perturbative calculations, it does not fit at all in the case of non-perturbative heat kernel-like methods.

Our second aim was to compute all the anomalies (trace and chiral) of Weyl, Dirac and Majorana fermions coupled to gauge potentials. The calculations are actually standard, but, once juxtaposed, they reveal a perhaps previously unremarked property: the chiral and trace anomalies due to a gauge background are rigidly linked.

**Author Contributions:** All author contribuite equally. All authors have read and agreed to the published version of the manuscript.

**Funding:** This research received no external funding.

**Acknowledgments:** S.Z. thanks SISSA for hospitality while this work was being completed. L.B. would like to thank his former collaborators A. Andrianov, M. Cvitan, P. Dominis-Prester, A. Duarte Pereira, S. Giaccari and T. Štemberga, with whom part of the material of this paper was elaborated or discussed.

**Conflicts of Interest:** The authors declare no conflict of interest

### Appendix A. Spin-States for Weyl Spinor Fields

In this appendix we aim to summarize the explicit form of the spin-states and plane wave functions for Weyl spinor fields in the more general and useful four component formalism. In order to realise a basis of spin states for a Weyl spinor field, we have to search among states of opposite chirality and frequency. Then, in so doing, we will be able to have at hand a complete and orthogonal quartet of spin-states for the 4D massless bispinor space. After setting $p^\mu = (|\mathbf{p}|, \mathbf{p}) = (\wp, p_x, p_y, p_z)$, we have

$$u_-(\mathbf{p}) = \frac{1}{\sqrt{\wp - p_z}} \begin{pmatrix} \wp - p_z \\ -p_x - ip_y \\ 0 \\ 0 \end{pmatrix}$$

The above spin-state is a positive frequency solution of the Weyl field equation and exhibits negative chirality: namely,

$$\not{p}\, u_-(\mathbf{p}) = 0 \qquad (\gamma_5 + 1)\, u_-(\mathbf{p}) = 0$$

Next we set

$$u_+(\mathbf{p}) = \frac{-1}{\sqrt{\wp + p_z}} \begin{pmatrix} 0 \\ 0 \\ \wp + p_z \\ p_x + ip_y \end{pmatrix}$$

which satisfies by construction

$$\not{p}\, u_+(\mathbf{p}) = 0 \qquad (\gamma_5 - 1)\, u_+(\mathbf{p}) = 0$$

Quite analogously, we can build up as well a pair of negative frequency spin-states of momentum $\mathbf{p}$ and both opposite chiralities: namely,

$$\not{\tilde{p}}\, v_\mp(\mathbf{p}) = 0 \qquad (\gamma_5 \pm 1)\, v_\mp(\mathbf{p}) = 0$$

where $\tilde{p}^\mu = p_\mu$. We find

$$v_-(\mathbf{p}) = \frac{1}{\sqrt{\wp - p_z}} \begin{pmatrix} p_x - ip_y \\ \wp - p_z \\ 0 \\ 0 \end{pmatrix}$$

together with

$$v_+(\mathbf{p}) = \frac{1}{\sqrt{\wp + p_z}} \begin{pmatrix} 0 \\ 0 \\ -p_x + ip_y \\ \wp + p_z \end{pmatrix}$$

It turns out that the above defined quartet of massless and chiral bispinor spin-states does fulfil orthogonality and closure relations. As a matter of fact, spin-states of opposite chirality and/or opposite frequency are orthogonal, as it does, and furthermore we get

$$(2\wp)^{-1}[\, u_\pm(\mathbf{p}) \otimes u_\pm^\dagger(\mathbf{p}) + v_\pm(\mathbf{p}) \otimes v_\pm^\dagger(\mathbf{p})\,] = \mathbb{P}_\pm$$

with of course $\mathbb{P}_{\pm} = \frac{1}{2}(\mathbb{I} \pm \gamma_5)$ which represent the closure or completeness relations for the massless and chiral bispinor spin-states basis quartet. The corresponding plane-wave functions, which are normal-mode solutions of the massless Dirac equation, read

$$\begin{cases} u_{\pm,\mathbf{p}}(x) = [(2\pi)^3 2\wp]^{-\frac{1}{2}} u_{\pm}(\mathbf{p}) \, e^{-i\wp t + i\mathbf{p}\cdot\mathbf{x}} \\[2mm] v_{\pm,\mathbf{p}}(x) = [(2\pi)^3 2\wp]^{-\frac{1}{2}} v_{\pm}(-\mathbf{p}) \, e^{i\wp t - i\mathbf{p}\cdot\mathbf{x}} \end{cases}$$

and fulfil in turn orthonormality and closure relation with respect to the usual Poincaré invariant inner product. It is crucial to gather, for a better understanding of the matter, that for any given frequency and chirality the spin-states and wave-functions of particle and anti-particle enjoy opposite wave vectors, i.e., opposite helicity.

**Appendix B. Consistent Gauge Anomaly with PV Regularisation**

To implement a PV regularisation, we replace $\widetilde{F}_{\mu\nu\lambda}^{(R)}(k_1, k_2)$ with

$$\begin{aligned} \widetilde{F}_{\mu\nu\lambda}^{(R)}(k_1, k_2) &= \int \frac{d^4 p}{(2\pi)^4} \, \mathrm{tr} \left\{ \frac{1}{\not{p} + m} \frac{1 - \gamma_5}{2} \gamma_\nu \frac{1}{\not{p} - \not{k}_1 + m} \frac{1 - \gamma_5}{2} \gamma_\lambda \frac{1}{\not{p} - \not{q} + m} \frac{1 - \gamma_5}{2} \gamma_\mu \right. \\ &\left. - \frac{1}{\not{p} + M} \frac{1 - \gamma_5}{2} \gamma_\nu \frac{1}{\not{p} - \not{k}_1 + M} \frac{1 - \gamma_5}{2} \gamma_\lambda \frac{1}{\not{p} - \not{q} + M} \frac{1 - \gamma_5}{2} \gamma_\mu \right\} \end{aligned} \quad \text{(A1)}$$

$m$ and $M$ are IR and UV regulators, respectively, and tr is the trace of gamma matrices.

Contracting with $q^\mu$ and working out the traces one gets

$$q^\mu \widetilde{F}_{\mu\nu\lambda}^{(R)}(k_1, k_2) = -2i\epsilon_{\mu\nu\rho\lambda} \int \frac{d^4 p}{(2\pi)^4} \left( 2p \cdot q \, p^\mu - p^2 q^\mu - q^2 p^\mu \right) (p - k_1)^\rho \left( \frac{1}{\Delta_{m^2}} - \frac{1}{\Delta_{M^2}} \right) \quad \text{(A2)}$$

where

$$\begin{aligned} \Delta_{m^2} &= (p^2 - m^2)((p - k_1)^2 - m^2)((p - q)^2 - m^2), \\ \Delta_{M^2} &= (p^2 - M^2)((p - k_1)^2 - M^2)((p - q)^2 - M^2) \end{aligned}$$

For later use we introduce also

$$\begin{aligned} \Omega_{m^2} &= ((p - k_1)^2 - m^2)((p - q)^2 - m^2), & \Lambda_{m^2} &= (p^2 - m^2)((p - k_1)^2 - m^2), & \text{(A3)} \\ \Omega_{M^2} &= ((p - k_1)^2 - M^2)((p - q)^2 - M^2), & \Lambda_{M^2} &= (p^2 - M^2)((p - k_1)^2 - M^2). & \text{(A4)} \end{aligned}$$

Now all the integrals are convergent because the divergent terms have been subtracted away. Let us proceed

$$\begin{aligned} q^\mu \widetilde{F}_{\mu\nu\lambda}^{(R)}(k_1, k_2) &= -2i\epsilon_{\mu\nu\rho\lambda} \int \frac{d^4 p}{(2\pi)^4} \left\{ \left[ \frac{-k_2^\mu (p - k_1)^\rho}{\Omega_{m^2} \Delta_{M^2}} + \frac{p^\mu k_1^\rho}{\Lambda_{m^2} \Delta_{M^2}} \right] \right. \\ &\quad \left( m^6 - M^6 + \left( M^4 - m^4 \right) \left( p^2 + (p - k_1)^2 + (p - q)^2 \right) \right. \\ &\quad \left. + \left( m^2 - M^2 \right) \left( (p - k_1)^2 (p - q)^2 + (p - k_1)^2 p^2 + p^2 (p - q)^2 \right) \right) \\ &\quad \left. + m^2 \left( p^\mu k_1^\rho - q^\mu (p - k_1)^\rho \right) \left( \frac{1}{\Delta_{m^2}} - \frac{1}{\Delta_{M^2}} \right) \right\} \end{aligned} \quad \text{(A5)}$$

The last line does not contribute, for the integrals converge (separately) and give a finite result, but since they are multiplied by $m^2$ they vanish in the limit $m \to 0$. Therefore, the last line can be dropped.

Now the strategy consists in simplifying separately each monomials in the numerator with a corresponding term in the denominator. For instance, if in a term of order $M^*$ there is the ratio $p^2/(p^2 - m^2)$, write $p^2$ as $p^2 - m^2 + m^2$. The $p^2 - m^2$ can be simplified with a corresponding term in the denominator. If $p^2 - m^2$ in the denominator is missing, there will be $p^2 - M^2$. We write $p^2$ as $p^2 - M^2 + M^2$, and $p^2 + M^2$ can be simplified, while the term proportional to $M^2$ remains and contributes to the term of order $M^{*+2}$. Proceed in the same way also with $(p-q)^2$ and $(p-k_1)^2$. Many terms (such as those of order $M^6$) cancel out. What remains is

$$q^\mu \widetilde{F}^{(R)}_{\mu\nu\lambda}(k_1, k_2) = -2i\epsilon_{\mu\nu\rho\lambda} \int \frac{d^4p}{(2\pi)^4} \left\{ \right. \tag{A6}$$

$$p^\mu k_1^\rho \left[ \frac{(M^2 - m^2)^2}{\Lambda_{m^2}\Lambda_{M^2}} - \frac{(M^2 - m^2)}{\Lambda_{M^2}} \left( \frac{1}{p^2 - m^2} + \frac{1}{(p-k_1)^2 - m^2} \right) - \frac{M^2 - m^2}{\Delta_{M^2}} \right]$$

$$-k_2^\mu(p - k_1)^\rho \left[ \frac{(M^2 - m^2)^2}{\Omega_{m^2}\Omega_{M^2}} - \frac{(M^2 - m^2)}{\Omega_{M^2}} \left( \frac{1}{(p-k_1)^2 - m^2} + \frac{1}{(p-q)^2 - m^2} \right) - \frac{M^2 - m^2}{\Delta_{M^2}} \right] \left. \right\}$$

It is easy too verify that, after introducing the relevant Feynman parameters, most of the terms vanish either because there is only one $p$ in the numerator or because of the anti-symmetry of the $\epsilon$ tensor. Only the last term in each line remains, so that:

$$q^\mu \widetilde{F}^{(R)}_{\mu\nu\lambda}(k_1, k_2) = 2i\, M^2 \epsilon_{\mu\nu\rho\lambda} \int \frac{d^4p}{(2\pi)^4} \frac{p^\mu k_1^\rho - k_2^\mu(p-k_1)^\rho}{\Delta_{M^2}} \tag{A7}$$

Next we introduce two Feynman parameters $x$ and $y$, shift $p$ like in Section 6.1 and make a Wick rotation on the momenta. Then, (A6) becomes

$$
\begin{aligned}
q^\mu \widetilde{F}^{(R)}_{\mu\nu\lambda}(k_1, k_2) &= -4\, M^2 \epsilon_{\mu\nu\rho\lambda} \int_0^1 dx \int_0^{1-x} dy \int \frac{d^4p}{(2\pi)^4} \frac{(1-x)k_1^\mu k_2^\rho}{(p^2 + M^2 + A(x,y))^3} \\
&= -\frac{1}{8\pi^2} M^2 \epsilon_{\mu\nu\rho\lambda} \int_0^1 dx \int_0^{1-x} dy \frac{(1-x)k_1^\mu k_2^\rho}{M^2 + A(x,y)} \\
&= -\frac{1}{24\pi^2} \epsilon_{\mu\nu\rho\lambda} k_1^\mu k_2^\rho
\end{aligned}
\tag{A8}
$$

Adding the cross term we get

$$q^\mu \widetilde{T}^{(R)}_{\mu\nu\lambda}(k_1, k_2) = -\frac{1}{12\pi^2} \epsilon_{\mu\nu\rho\lambda} k_1^\mu k_2^\rho \tag{A9}$$

which is the same result as in Section 6.1.

## Appendix C. Consistent and Covariant Gauge Anomalies

In this Appendix we briefly recall the difference between covariant and consistent anomalies. The reason why (87) is called covariant is self-evident, whereas why (73) is called consistent is not so straightforward.

Let us consider a generic gauge theory, with connection $V_\mu^a T^a$, valued in a Lie algebra $\mathfrak{g}$ with anti-Hermitean generators $T^a$, such that $[T^a, T^b] = f^{abc}T^c$. In the following it is convenient to use the more compact form notation and represent the connection as a one-form $\mathbf{V} = A_\mu^a T^a dx^\mu$, so that the gauge transformation becomes

$$\delta_\lambda \mathbf{V} = \mathbf{d}\lambda + [\mathbf{V}, \lambda] \tag{A10}$$

with $\lambda(x) = \lambda^a(x)T^a$ and $\mathbf{d} = dx^\mu \frac{\partial}{\partial x^\mu}$. The mathematical problem is better formulated if we promote the gauge parameter $\lambda$ to an anticommuting field $c = c^a T$, the Faddeev–Popov ghost, and define the BRST transformation as

$$s\mathbf{V} = \mathbf{d}c + [\mathbf{V}, c], \qquad sc = -\frac{1}{2}[c, c] \tag{A11}$$

The operation $s$ is nilpotent, $s^2 = 0$. We represent with the same symbol $s$ the corresponding functional operator, i.e.,

$$s = \int d^d x \left( s\mathbf{V}^a(x) \frac{\partial}{\partial \mathbf{V}^a(x)} + sc^a(x) \frac{\partial}{\partial c^a(x)} \right) \tag{A12}$$

The Ward identity for the gauge (BRST) transformation is classically given by $s\,W[V] = 0$, but at the first quantum level one may find an anomaly

$$s\,W[V] = \hbar\,\mathcal{A}[V, c] \tag{A13}$$

the RHS is an integrated expression linear in $c$, and once the latter is removed, in 4D it becomes precisely the RHS of (73).

Due to the nilpotency of $s$ the anomaly must satisfy

$$s\mathcal{A}[V, c] = 0 \tag{A14}$$

These are the famous Wess–Zumino consistency conditions (it is enough to functionally differentiate (A14) by $c^a$ to cast them in the original form). $\mathcal{A}[V, c]$ is a cocycle of the cohomology defined by $s$. If there exists a local expression $\mathcal{C}[V]$ such that $\mathcal{A}[V, c] = s\mathcal{C}[V]$, the cocycle is a coboundary (trivial anomaly). If such $\mathcal{C}[V]$ does not exist, we have a true anomaly.

An easy way to verify the consistency conditions is via the descent equations. To construct the descent equations we start from a symmetric polynomial in the Lie algebra of order 3, $P_3(T^a, T^b, T^c)$, invariant under the adjoint transformation:

$$P_3([X, T^a], T^b, T^c) + \ldots + P_3(T^a, T^b, [X, T^c]) = 0 \tag{A15}$$

for any element $X$ of the Lie algebra. In many cases these polynomials are symmetric traces of the generators in the corresponding representation

$$P_3(T^a, T^b, T^c) = Str(T^a T^b T^c) = d^{abc} \tag{A16}$$

($Str$ denotes the symmetric trace). With this one can construct the 6-form

$$\Delta_6(\mathbf{V}) = P_n(\mathbf{F}, \mathbf{F}, \ldots \mathbf{F}) \tag{A17}$$

where $\mathbf{F} = \mathbf{d}\mathbf{V} + \frac{1}{2}[\mathbf{V}, \mathbf{V}]$. It is easy to prove that

$$P_3(\mathbf{F}, \mathbf{F}, \ldots \mathbf{F}) = \mathbf{d}\left( n \int_0^1 dt\, P_n(\mathbf{V}, \mathbf{F}_t, \ldots, \mathbf{F}_t) \right) = \mathbf{d}\Delta_{2n-1}^{(0)}(\mathbf{V}) \tag{A18}$$

where we have introduced the symbols $\mathbf{V}_t = t\mathbf{V}$ and its curvature $\mathbf{F}_t = \mathbf{d}\mathbf{V}_t + \frac{1}{2}[\mathbf{V}_t, \mathbf{V}_t]$, where $0 \leq t \leq 1$. In the above expressions the product of forms is understood to be the exterior product. To prove

Equation (A18) one uses in an essential way the symmetry of $P_3$ and the graded commutativity of the exterior product of forms. Equation (A18) is the first of a sequence of equations that can be proven

$$\Delta_6(\mathbf{V}) - \mathbf{d}\Delta_5^{(0)}(\mathbf{V}) = 0 \tag{A19}$$

$$s\Delta_5^{(0)}(\mathbf{V}) - \mathbf{d}\Delta_4^{(1)}(\mathbf{V}, c) = 0 \tag{A20}$$

$$s\Delta_4^{(1)}(\mathbf{V}, c) - \mathbf{d}\Delta_3^{(2)}(\mathbf{V}, c) = 0 \tag{A21}$$

$$\cdots\cdots$$

$$s\Delta_0^{(5)}(c) = 0 \tag{A22}$$

All the expressions $\Delta_k^{(p)}(V, c)$ are polynomials of $\mathbf{d}, c$ and $\mathbf{V}$. The lower index $k$ is the form order and the upper one $p$ is the ghost number, i.e., the number of $c$ factors. The last polynomial $\Delta_0^{(5)}(c)$ is a 0-form and clearly a function only of $c$. All these polynomials have explicit compact form. They were originally introduced by S.S. Chern [20], in the geometric context of a principal fibre bundle, which gives them full meaning. Here we limit ourselves to a formal application, which however can be proven to lead to correct results.

The RHS contains the general expression of the consistent gauge anomaly in 4 dimensions, for, integrating (3.53) over space-time, one gets

$$
\begin{aligned}
s\,\mathcal{A}[c, \mathbf{V}] &= 0 \tag{A23}\\
\mathcal{A}[c, \mathbf{V}] &= \int d^d x\, \Delta_4^1(c, \mathbf{V}), \qquad \text{where}\\
\Delta_4^1(c, \mathbf{V}) &= 6 \int_0^1 dt (1-t) P_3(\mathbf{d}c, \mathbf{V}, \mathbf{F}_t)
\end{aligned}
$$

One can verify that $\mathcal{A}[c, \mathbf{V}]$ is the anomaly (73), saturated with $c^a$ and integrated over space-time, up to an overall numerical coefficient.

To verify that the anomaly (73) is non-trivial one can use a *reductio ad absurdum* argument. Let us suppose that (A23) is trivial. Then using repeatedly the local Poincaré lemma one can prove that there must exist a 0-form $C_0^{(2n-2)}(c)$ such that

$$\Delta_0^{(5)}(c) = s\, C_0^{(4)}(c) \tag{A24}$$

However this is impossible, for the expression for $\Delta_0^{(5)}(c)$ is

$$\Delta_0^{(5)}(c) \sim P_n(c, [c, c]_+, \ldots, [c, c]_+) \tag{A25}$$

and the only possibility for $C_0^{(4)}(c)$ to satisfy (A24) is to have the form

$$C_0^{(4)}(c) \sim P_n(c, c, [c, c]_+) \tag{A26}$$

which, however, vanishes due to the symmetry of $P_3$ and the anticommutativity of $c$.

It is evident that in the Abelian case the above construction becomes trivial, and consistent and covariant gauge anomalies collapse to the same expressions. The only difference is the numerical factor in front of them.

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
