# Peer review of "Dirac, Majorana, Weyl in 4D"

_universe, doi:10.3390/universe6080111_

Round 1

Reviewer 1 Report

The manuscript presents an enlightening and pedagogically written review of subtle issues distinguishing definitions, action, and anomalies for Weyl, Dirac and Majorana fermions coupled to gauge potentials. Although the relevant literature is vast, many of these aspects are subtle and are sometimes misunderstood. Bringing them together and, in particular, accompanying the reader through the explicit Feynman diagram calculations of the anomalies is a useful contribution to the literature.

The topic of this review is interesting, the results and discussions are technically solid and very detailed. The exposition is clear and pedagogical, combining discussions of fundamental issues and practical calculations. In short, this review will be a very valuable text especially for students and young researchers who wish to master the formal aspects doing QFT with massless fermions. It certainly deserves publication in Universe.

One request I have is to express opinion on others' publications or notes on similar issues. For instance, Howard Haber wrote some time ago a short note http://scipp.ucsc.edu/~haber/webpage/majnu.pdf with the title "Massless Majorana and Weyl fermions cannot be distinguished" which sounds contradicting to the present manuscript. What he meant there was that a theory with a massless fermion can be described either way, at least at the level of lagrangian, and there is no preferred formulation. Since he present manuscript goes beyond just writing down lagrangians, the authors may have comments on it.

Author Response

We thank the referee for the report.

Concerning the question raised at the end of it, we think the answer is rightly guessed by the referee her/himself. Anyhow it is as follows: it is well-known that many aspects of a theory of a Weyl fermion and a massless Majorana fermion may overlap, but of course not all of them do. There have been also attempts to describe to the standard model only in terms of Majorana fermions. It is also a common practice in supersymmetric model building, for instance, to use indifferently Weyl or Majorana fermions. This is perfectly allowed. However once the model is constructed one must specify what are the true physical degrees of freedom. But it is enough to think of the S-matrix and the reduction formulas where spin states are necessary: the spin states for Weyl and Majorana are completely different (see the new section 5 and the new Appendix A)! The lore that the standard model, for one, can be expressed in terms of Majorana fermions is a `legend' that persists and produces enormous damages. In any case, when a theory is expressible in two different languages, a precaution clause wants it that one develops the two formalisms independently and eventually compare the results. This is what we do in our paper, we compute anomalies for Weyl and Majorana fermions independently and, obviously since chirality is crucial in this type of calculations, we find different results.

Reviewer 2 Report

Please, find my report in the attached pdf file.

Author Response

\documentclass[10pt]{article}

\begin{document}

%

\begin{center}

{\Large\textit{Reply to the referee report on}

\\

{\Large Dirac, Majorana, Weyl in 4d}}

\bigskip

{\large by L. Bonora, R. Soldati, S. Zalel}

\end{center}

\noindent

This referee report is an embarassing collection of the most gigantic errors that can be met in the literature concerning spinors. It is amazing that in the advanced XXI century there are still physicists who believe and even speak out such enormities as the mass of a Weyl fermion or the identity between Weyl and Majorana fermions. In a period when the very nature of neutrinos and how they should be mathematically represented is under extremely challenging experimental scrutiny, such enormities are inadmissible (why do people speak of the alternative between Dirac and Majorana neutrino, and not between Dirac and Weyl?). We are outraged by the fact that this report has not been excluded at the very beginning from the refereeing process. There is a limit of decency that should not be passed.

But, we must say, there is also a funny aspect in it. Even supposing that a theory is expressible in two different languages (Weyl and Majorana), a precaution clause wants it that one develops the two formalisms independently and eventually compare the results. This is what we do in our paper, we compute anomalies for Weyl and Majorana fermions independently and, obviously, since chirality is crucial in this type of calculations, we find different results. But the referee wants to forbid us do that. No, it is not allowed to treat Weyl fermions as Weyl fermions, one is allowed only to treat them as Majorana fermions! On the face of it what can one say?

The only positive aspect in this desolation is that the other four reports are in total disagreement with this one. It seems crackpots are not in power yet!

As we are requested by the editor's letter, we patiently repeat once again what is already in the text of our paper.

Things are since long very well and firmly established. A classical, e.g. left, Weyl spinor is a 2-component complex or Grassmann valued

on the Minkowski space, which transforms according to an IRREP representation $ D(\frac12,0) $ of the Lorentz group: namely,

\begin{eqnarray}

\psi_{L}(x)=\left\lgroup

\begin{array}{c}

\psi_{L1}(x)

\\

\psi_{L2}(x)

\end{array}

\right\rgroup

\qquad\quad

\psi_{L}'(x')=\exp \left\lbrace\textstyle\frac12i\vec{\sigma}\cdot(\vec{\alpha}-i\vec{\eta}\,)\right\rbrace\,\psi_{L}(x)

\end{eqnarray}

where $ \vec{\alpha},\vec{\eta} $ are angular and rapidity canonical coordinates on the Lorentz group manifold.

The left Weyl spinor can be as well represented in the 4-component bispinor form, by projecting a Dirac spinor $ \psi $, viz.,

$$\psi_{L}(x)=P_{L}\psi(x)=\textstyle\frac12(1-\gamma_{5})\psi(x)$$ the correspondence between the two and four component forms

being evident from the nature of the left projector onto a 2d space, what justifies the use of the same symbol to denote both.

Notice, however, that the parity and charge conjugation discrete transformations, for example, can be implemented for Weyl

spinors \textbf{only} in the 4-component form, since both chiralities, left and right, are actually involved. In the 4-component

formalism, left and right Weyl spinors are eigenstates of the chiral matrix $ \gamma_{5} $ with eigenvalues $ (-1) $ and $ (+1) $

respectively.

A Majorana spinor is a \textbf{self-conjugated bispinor}, which can be build out of a Weyl spinor: for instance, in the Weyl representation

of the Dirac matrices we can quite generally write

\begin{eqnarray}

\Psi_{L}(x)=\left\lgroup

\begin{array}{c}

\psi_{L}(x)

\\

0

\end{array}

\right\rgroup

\qquad\quad

\Psi_{M}(x)=\left\lgroup

\begin{array}{c}

\psi_{L}(x)

\\

-\sigma_{2}\psi_{L}^{\ast}(x)

\end{array}

\right\rgroup

\label{spinoriWM}

\end{eqnarray}

A Majorana bispinor transforms according to the reducible 4d representation $ D(\frac12,0)\oplus D(0,\frac12) $

of the Lorentz group. Moreover, it is an eigenstate of the charge conjugation transform $ \psi^{c}=\gamma^{2}\psi^{\ast} $

with eigenvalue $ (+1) $ for $ \Psi_{M}^{c}=\Psi_{M} $.

From the above simple, explicit and quite general form (\ref{spinoriWM}) it is quite evident that \textsc{Weyl and Majorana spinor can

never be identified} and there is no invertible 1:1 map that could connect them. The Weyl spinors exhibit only one chirality, while

the Majorana spinors necessarily contains both kinds of chirality. Since the mass term necessarily involves both kinds of

chirality, owing to Poincar\'e invariance, it is absolutely evident that Dirac and Majorana bispinors can be massive, while

\textsc{a Weyl spinor is necessarily massless}.

Any left Weyl spinor wave or quantum state of definite (positive) frequency exhibits only negative helicity, while any massless

or massive Majorana spinor wave or quantum state of definite (positive) frequency exhibits both kinds of helicity.

Any left Weyl spinor quantum particle admits its antiparticle of the same energy and chirality, though opposite momentum and helicity.

Owing to its complex nature, the Weyl spinor dynamics is invariant with respect to U(1) phase transformations. Then a conserved charge

always exists, which is usually called lepton number for historical reasons, which is opposite for particles and antiparticles.

It has been showm by Ettore Majorana - \textit{Teoria simmetrica dell'elettrone e del positrone}, Il Nuovo Cimento,

volume \textbf{14} (1937) pp. 171-184 - that a purely imaginary representation for the Dirac matrices exists, in which

the Majorana bispinor can always be chosen to be real. As a consequence, always due to Poincar\'e covariance, the massive or massless

Majorana bispinor is neutral, it does

not carry any lepton number and neither does it admit any distinguished antiparticle.

\medskip

It appears quite evident from the above listed basic, very well-known, long standing issues, that the anonymous referee is far away from being able

to write a report on the present subject, his incompetence being so much surprisingly and embarassingly total and absolute.

We believe it is useless to argue with her/him and her/his unshakable certainties. How can one maintain, as s/he does, that from a Majorana fermion theory one can extract chiral information when in the initial theory chirality is not even defined? This is the most ludicrous one among various grotesque pleasantries of the report. This referee is completely outside any acceptable standard of knowledge on quantum field theory and knows absolutely nothing about spinor properties and features.

\medskip

It is perhaps more interesting to wonder why the above pleasantries are not infrequent in the literature, although not often expressed in the extreme fideistic manner of the report. It is true that some aspects of a Weyl fermion theory can be expressed by means of Majorana fermions. It is also true that some attempts have been made to express the standard model in terms of Majorana fermions, but only {\it some} aspects! Think of the reduction formulas, where spin states are needed and are different for Weyl and Majorana fermions? The formulation of the standard model by means of Majorana fermions is another popular `legend' that contributes only to confusion. Why and where such distorted knowledge originates we can only formulate hypotheses:

1) from some careless statements from leading physicists, taken too literally by less spirited people.

2) from a confusion among dimensions: a large amount of the literature on fermions concerns 2 and 10 dimensions, where Weyl-Majorana fermions exist; of course this does not hold in 4d.

3) from the model construction of supersymmetric theory, which makes often use of Weyl and Majorana fermions indifferently; but of course one thing is the model building, quite another thing the physical use of the models, in which one has to specify what are the true physical degrees of freedom.

This and the fervid imagination of our referee can perhaps explain the gallery of horrors of the report.

\vfill\eject

\end{document}

Reviewer 3 Report

The manuscript ``Dirac, Majorana, Weyl in 4d” by Bonora et al. reviewed the physics of Weyl fermion and massless Dirac, Majorana fermions in four dimensional flat spacetime,  focusing on aspects of the low energy effective action of Weyl fermions and chiral and trace anomalies of Weyl, Dirac, Majorana fermions coupled to gauge potential.

The techniques used in the manuscript are standard in quantum field theory and should be solid. The physical results reviewed here are the most important aspects of Dirac, Majorana, Weyl fermions.  I recommend this manuscript to be published in Universe after the authors consider the improvements on the manuscript according to the following comments.  

(1) In line 77, What is $\alpha^2$?

(2) The authors are suggested to use a table to summarize the properties of equations (9)-(16) to make the content clear to the readers.  

(3) A question to equation 19: Since \psi is a Majorana spinor, we have \psi=\hat{\psi}, why are both \psi and \hat{\psi} used in equation 19?   

(4) In section 2.1, it might be better to write out explicitly the action for massless Majorana fermion in the chiral representation of gamma matrices to show that it takes the ``same" form as Weyl fermion as stated in line 175 since Majorana fermions are widely discussed in Majorana representations of gamma matrices in the literature.  

(5) The authors need to specify $\alpha^\nu$ in equation 31.  

(6) In section 4, the authors discussed the effective action of Weyl fermions by adding the non-interacting Lagrangian $\mathcal{L}’$ below line 327. Can the authors briefly comment on the radiative correction to the kinetic term of (massless) Majorana fermions?   

(7) In line 509, ``eq. (109)" should be a typo.  

(8) The link between the chiral anomaly and trace anomaly is quite interesting. The symmetries associated to these anomalies are different. Can the authors briefly comment on the reason why there is such an interesting link. 

Author Response

We thank the referee for report and suggestions. Concerning the points raised.

(1,5,7) the text has been changed accordingly

(2) a table has been introduced as suggested by the referee.

(3) $\hat \psi$ is used, instead of $\psi$ for convenience, in order to derive the subsequent equations.

(4) the suggestion has been accepted, see the new section 5.

(6) in the case of massless Majorana fermion there are no radiative corrections because it does not minimally couple to a the electromagnetic field (it is `sterile').

(8) The link between chiral and trace anomalies has been discovered in the framework of supersymmetric theories were both belong to the same supermultiplet. In our paper we point out that this relation is more general and valid for any gauge theory (a similar relation exists in gravity theories too, but this is beyond the scope of the present paper). At present we do not have an explanation beyond the fact that the relevant diagrams are very similar and the tiny difference is actually vanishing. We think a full explanation should come form non-perturbative methods (we are pursuing this direction).

Reviewer 4 Report

In this review, the authors introduce the and highlight the differences of Dirac, Weyl, and Majorana fermions in 4D Minkowski spacetime. They pay particular detail to their mutual differences (as concerns definitions and properties) and clarify several common misconceptions. They then discuss the nature of quantum anomalies for Weyl and Majorana fermions: For this prupose, they introduce the functional integral for relativistic fermions and discuss several conceptual and computational issues. For the computation of anomalies, the authors use the explicit pertrubative formalism. They compute the chiral and trace anomalies and show how they are linked. The latter result appears to be new, or at least not widely acknowledged in the literature.

I think this review will be very helpful for researcher in the field and I found the content useful. The exposition is rather brief in the beginning, so I would expect the audience to have some prior knowledge on relativistic fermions and the basic notation. However, given that there are many lengthy introductions to this topic in textbooks, I find the short version helpful, because it quickyl gets the reader to the actual results. The review is sufficiently well-written and for most parts has a clear line of thought. Below, I am giving some recommendations on those parts where the presentation should be improved. I recommend the review for publication once my remarks below have been incorporated.

The following points should be addressed in a revised version of the manuscript:

(1) "Section 6" is missing in the overview at the bottom of page 2.
(2) Section on Notation: The matrices \alpha^\nu should be defined, since they are also used below in Eq. (31).
(3) Define \bar{\psi}.
(4) Above Eq. (4): The definition of (\Psi_1,\Psi_2) is incomplete, since the right-hand side <...|...> is not defined either. Rather this anticipates that every reader would understand <...|...> ad the standard scalar product, which, in this context, is certainly not right. Rather define \Psi as a four-component column vector and then define (\Psi_1,\Psi_2) = \Psi_1^\dagger \gamma^0 \Psi_2 as the usual matrix product.
(5) Hats over \psi_R and \psi_L are inconsistent: Sometimes they are written as hat, sometimes as widehat.
(6) The presentation of C, P, CP transformationf of the Weyl action is in the wrong order. A more logical approach, in my opinion, would be to first present (or better derive) Eq. (16). This implies Eqs. (11) and (13), where both C and P merely exchange L and R. This then immediately implies that CP leaves the action invariant.
(7) Below Eq. (22): Why does this have the same form "as a massless Majorana action". The latter has not been defined above in two-component notation involving the Pauli matrices. This should be clarified.
(8) Below Eq. (32): Why does it follow that the operator \omega_L "is singular and does not possess any rank-four inverse"?
(9) Eq. (35): Replace g_{\mu\nu} be \eta_{\mu\nu}?
(10) Before Eq. (37). It should be stated that the first calculation is done in "dimensional regularization", since the phrase is used below.
(11) Eq. (46): Explain "+ evansecent"
(12) Below Eq. (81): "This is nothing but (74)." Do you mean (64) instead?
(13) It would be helpful to define "consistent" and "covariant" anomaly more prominently. Right now this aspect is somethwta hidden between equations.
(14) Section 8, first paragraph: Spell out "energy-momentum tensor" (could be confused with electromagnetic tensor)
(15) Eq. (108): Should this be <<T_{5\mu}^\mu>>?
(16) At last, is the "strict" link between anomalies exact beyond perturbation theory?

Author Response

We thank the referee for the report and some useful suggestions.

Concerning the points raised in the report: we thank the referee for signalling several typos, (1,2,3,9,12,14), which have been corrected.

(4) the part in question has been suppressed as inessential.

(5) we have added a footnote unambiguously explaining the meaning of barred and hatted quantities.

(6) we do not quite understand the referee's point. Since it is expressed as an opinion we take the liberty to insist on our exposition.

(7) the referee's suggestion has been inserted in a new section 5, devoted to quantum Majorana fermions.

(8) the operator w_L is not invertible because it contains a projector P_L.

(10) The specification has been added.

(11) 'Evanescent' is a common terminology in field theory meaning a term that vanishes when the regulator is removed.

(13) We have added Appendix B, where the adjectives `consistent' and `covariant' are defined and explained.

(15) No, eq.(108) is correct as it is. According to our method we use the divergence of the chiral current to compute the trace anomaly.

(16) We believe the link between trace and chiral anomaly persists non-perturbatively, a preliminary calculation says so. On the other hand, anomalies in FT are considered
non-perturbative phenomena, often of topological nature. It would be surprising that the link exists only perturbatively. However we would like to make more checks on the anomalies' coefficients before making a definitive claim.

Reviewer 5 Report

The authors review the differences between Dirac, Weyl and Majorana spinors in 4D focusing in particular in the massless case.
The review gathers in one place many interesting, albeit not new, results that may interest a wider audience.

A few questions need to be clarified:

1. After eq.(89) it is stated that for a Majorana field J_R and J_L are related but the explicit relation is not given. In fact, the definition in eq.(87) implies that J_L=-J_R and the vector current in eq.(88) seems to vanish identically. Is this indeed so? Or does quantum corrections intervene?

2. For the calculations of the triangle diagrams such as in (53), which are superficially linearly divergent, one usually introduces arbitrary offset 4-vectors to deal with the ambiguity in the regularization procedure. Is this freedom absent in the calculations?

3. In Sec.3, the authors start to discuss that the functional measures for the massless Weyl and Majorana spinors are substantially different and they proceed to discuss functional determinants.
They point out that the naive kinetic opeartor for a Weyl spinor is not invertible but do not discuss the difference to the Majorana case.
The discussion should be extended to include the Majorana case.

Other minor questions and suggestions follow.

4. After eq.(1), it should be specified that C=gamma^0 gamma^2 is only valid in certain representations such as Dirac or Weyl, it is not valid for example in the Majorana representation which is later used briefly.

5. For pedagogical reasons, I believe the plane wave expansion analogous to (17) should be shown for a Majorana field. The expansion may illustrate the difference between Weyl and Majorana, even for the massless case.

6. In sec.3, second paragraph, "Dirac operator proper" is not grammatical and probably involves a typo.

7. Comma needed in the displayed equation after (47).

8. Typo "normalzation" after eq.(65). A careful reading is advised to check for other possible typos.

Author Response

We thank the referee for report and suggestions. Concerning the questions raised:

1. The referee is right, in fact $J_L=-J_R$ and $J_M=0$, as it should for a Majorana fermion. We have added a couple of lines and suppressed two.

2. Our prescription for dimensional regularisation, as one can gather from the formulas, is the Breitenlohner-Maison one: first regulate the integral, then and only then move gamma-matrices and compute traces; finally evaluate integrals by introducing Feynman parameters. This prescription is totally unambiguous and yields the same results
as the PV regularization. Actually, to our knowledge, it is the only one which is fully self-consistent from the algebraic and analytic points of view. Other prescriptions, like for instance the rightmost $\gamma_5$, lead sooner or later to inconsistencies.

3. We have accepted this suggestion by introducing a new section 5, which answers also point 5.

6,7,8. The text has been changed accordingly.